

# An evaluation of airborne mass balance and tracer correlation approaches to estimate site-level CH$_4$ emissions from LNG facilities using CO$_2$ as a tracer of opportunity

Mark F. Lunt[1], Stephen J. Harris[2,3], Jorg Hacker[4,5], Ian Joynes[6], Tim Robertson[7], Simon Thompson[7], James L. France[8,9]

[1]Environmental Defense Fund, Perth, WA, Australia
[2]UNEP's International Methane Emissions Observatory (IMEO), Paris, France
[3]School of Biological, Earth and Environmental Sciences, University of New South Wales, Sydney, NSW, Australia
[4]Airborne Research Australia, Parafield, SA, Australia
[5]College of Science and Engineering, Flinders University, Adelaide, SA, Australia
[6]Woodside Energy Limited, Perth, WA, Australia
[7]Chevron Australia Pty Ltd, Perth, WA, Australia.
[8]Environmental Defense Fund Europe, Brussels, Belgium
[9]Earth Sciences Dept, Royal Holloway University of London, Egham, UK

*Correspondence to*: Mark Lunt (mlunt@edf.org)

**Abstract**

Accurate and representative quantification of methane (CH$_4$) emissions from individual oil and gas facilities is crucial to improve our knowledge of CH$_4$ sources, improve the reliability of emissions reporting and help facilitate mitigation opportunities. Liquefied Natural Gas (LNG) terminals primarily emit carbon dioxide (CO$_2$) but are also potentially large CH$_4$ sources in the gas supply chain. Here, in work supported by the United Nations Environment Program International Methane Emissions Observatory (UNEP's IMEO), we evaluate two airborne measurement approaches to quantify CH$_4$ emissions from four LNG facilities. The first approach applies a downwind mass balance method to quantify emissions of both CH$_4$ and CO$_2$. Since operator-reported CO$_2$ emissions are relatively well-established, we evaluate the method's performance by comparing to operator-reported CO$_2$ values. Using this approach, we show that an individual facility CO$_2$ mass balance quantification has a mean relative difference to operator reporting of $\pm 20\%$, with no significant mean bias across all estimates. The second approach uses measured CH$_4$:CO$_2$ mole fraction ratios as an alternative method for estimating site-level CH$_4$ emissions. Using this tracer correlation approach, we show that uncertainties in the ratio of CH$_4$:CO$_2$ from a single day can be below $\pm 10\%$ at the 95% confidence level. Due to uncertainty in the CO$_2$ emission rate, the resulting mean 2$\sigma$ uncertainty on CH$_4$ emissions is $\pm 30\%$. CH$_4$:CO$_2$ ratios are found to vary with height, with larger variation at lower altitudes ($< 250$m).





Representative sampling across both horizontal and vertical space is needed to enhance the accuracy of the tracer correlation approach for individual emission estimates. We find that when repeated over multiple days and different atmospheric conditions, ratio measurements below 250 m provide a median estimate that is within 5% of the mass balance and multi-height ratio methods. Whilst the $CO_2$:$CH_4$ emission ratios derived from measurements could be applied to estimate $CH_4$ emissions over longer timeframes, the degree of representativeness will depend on the variability of the ratio over time. Our results indicate that the tracer correlation approach using $CO_2$ as a tracer of opportunity can be used to efficiently estimate $CH_4$ emissions from LNG facilities with a low level of uncertainty.

## 1 Introduction

Methane ($CH_4$) is an abundant greenhouse gas, with over 80 times the global warming potential of carbon dioxide ($CO_2$) when evaluated over a 20 year time scale (Myhre et al., 2013). The Oil and Gas (O&G) sector has been estimated to account for 50-70 Tg yr$^{-1}$, roughly 25% of global anthropogenic $CH_4$ emissions (Saunois et al., 2024; Shen et al., 2023). A unique component of the gas supply chain are the emissions associated with the liquefaction of natural gas into Liquefied Natural Gas (LNG) for long-distance ocean transport. This process generates $CO_2$ emissions from multiple sources including gas turbines, venting and flares, in addition to $CH_4$ emissions (Balcombe et al., 2017).

Accurate knowledge of the extent of $CH_4$ emissions from the O&G sector relies heavily on atmospheric measurement studies that determine emission rates via measurements of enhancements in atmospheric $CH_4$ mole fractions emanating from the sources in question. In particular, studies that focus on individual facilities, termed site-level measurements, can be used to improve our knowledge of the distribution of emission sources, verify emissions reporting and identify mitigation opportunities (e.g. Stavropoulou et al., 2023; Zavala-Araiza et al., 2021). Site-level measurement and quantification also forms the basis for the highest level of reporting under the Oil and Gas Methane Partnership (OGMP) 2.0, where site-level measurements are used to cross-check emissions reported using more granular methods.

One common example of this type of site-level measurement is in-situ monitoring conducted using airborne platforms. Typically, aircraft are flown around or downwind of the facility and emissions inferred using a mass balance approach that quantifies the rate of emissions required to explain the difference in mole fractions measured



upwind and downwind of the facility. This approach has been applied to quantify $CH_4$ emissions from various parts of the O&G supply chain including onshore and offshore production (Foulds et al., 2022; Gorchov Negron et al., 2020; Lavoie et al., 2015; Li et al., 2024; Pühl et al., 2024).

The quantification accuracy of airborne mass balance approaches has commonly been assessed in two ways. The first is through controlled release experiments, whereby methane is released at a known location and rate and the resulting airborne measurement-based results compared to this known release rate (e.g. El Abbadi et al., 2024; Ravikumar et al., 2019; Sherwin et al., 2021). In this case, the release rate is known to a high degree of certainty.

However, the release setup may be idealised, consisting only of a single point source and other factors that simplify the quantification assessment process. Whilst this allows direct comparison to a known emissions rate, performance statistics may not be representative of deployment around more complicated or dispersed sources in the field. A second approach is to compare to some existing source of another tracer gas, where emission rates should be known, and use this to infer the likely accuracy for quantification of other sources. One example is the evaluation of $CO_2$

emission estimates against reported emissions from large sources such as power stations, where emissions rates of $CO_2$ should be well-known based on fuel consumption statistics (Hajny et al., 2023). This validation can then be used to infer the likely accuracy of quantification applied to other sources (Zhang et al., 2023).

LNG liquefaction facilities emit large quantities of $CO_2$, as well as $CH_4$ emissions. The $CO_2$ emitted originates from a variety of sources but predominantly from gas turbines used for liquefaction and generating on-site power, with an estimated 8-12% gas throughput used for onsite energy demand (Balcombe et al., 2017). Acid gas removal of reservoir $CO_2$ prior to liquefaction, and flaring are other significant sources dependent on the source gas reservoir $CO_2$ content and operating status. Zhang et al. (2023) identified 8 $CO_2$ point sources related to fuel and flare

combustion from an LNG facility using a remote sensing instrument, highlighting the multifaceted nature of $CO_2$ emissions. A variety of sources can contribute to $CO_2$ and $CH_4$ emissions but not all coincide in location and rate. Innocenti et al. (2023) found that between 50-90% of $CH_4$ emissions came from a combination of LNG trains, power generation and flares at three liquefaction sites. Additional $CH_4$ emissions can come from elements such as the boil off gas system and LNG storage, which may not be associated with $CO_2$ emissions. Given the expected

emissions of both gases, liquefaction facilities provide an opportunity to evaluate the quantification accuracy of $CO_2$ emissions, and thereby $CH_4$, at the same time as estimating the site-level $CH_4$ emissions sampled under the exact same conditions.



The airborne mass balance method is not the only approach to estimate site-level $CH_4$ emissions. An alternative method relies on measuring the mole fraction ratios of $CH_4$ relative to some tracer gas which is released at a known rate. This approach is commonly referred to in the literature as a tracer correlation, tracer-tracer, tracer flux ratio or similar (e.g. Delre et al., 2022; Roscioli et al., 2015; Zavala-Araiza et al., 2018). Under the ideal scenario of two co-located emissions, the ratio of measured mole fractions at any point in the downwind outflow should be equivalent to the ratio of emissions of the two gases. Provided the emission rate of the co-tracer is known and the emission rate of the tracers is constant throughout the duration of measurements, one can easily calculate the emission rate of methane. The advantage of this approach is that assumptions relating to the downwind transport of $CH_4$ are removed, since both gases should be transported the same way in the atmosphere. Atmospheric transport uncertainties are difficult to quantify but represent one of the major sources of uncertainty in aerial mass balance estimates since variable wind speed, wind direction and vertical mixing can mean the mass flow rate through the downwind plane is not the same as the emissions rate. The tracer correlation method removes this source of uncertainty but introduces additional uncertainties, particularly if the emissions of $CH_4$ and co-tracer are not completely co-located, or are emitted at different exhaust temperatures.

Measurements of the $CH_4$-tracer ratios are commonly performed at ground level, often from a mobile measurement platform (e.g. Delre et al., 2022; Roscioli et al., 2018; Yacovitch et al., 2017; Zavala-Araiza et al., 2018). The average uncertainty of a quantification using this ground-based mobile approach has been estimated to be around ±20% at the 95% confidence level (Delre et al., 2022). The tracer correlation method has also been applied using aerial measurements to estimate emissions from dairy farms (Daube et al., 2019). This study estimated whole farm $CH_4$ emissions to ±11% at the 95% confidence level using an aerial tracer correlation approach, but noted the approach might be improved by flying downwind transects up to 5 km downwind, rather than circles in close proximity, as well as using more than two point source releases for the tracer gas. Although LNG, or even O&G facilities are not directly comparable to dairy farms, the study results suggest there may be additional benefit from performing the tracer correlation method using aerial measurements over ground based.

These previous examples rely on having access to the target site and releasing the tracer at a known rate from as close to the $CH_4$ source as feasible. This is not always possible. However, given the relative size of emissions of $CO_2$ from LNG liquefaction facilities, the expected accuracy of emission factor estimates of $CO_2$ emissions, and





the proximity of these $CO_2$ sources with $CH_4$ sources, it stands to reason that $CO_2$ has the potential to be used as a tracer of opportunity to calculate $CH_4$ emissions from liquefaction sites.


As part of a study funded by the United Nations Environment Program's International Methane Emissions Observatory (UNEP's IMEO), aerial methane measurement surveys of four LNG liquefaction facilities were undertaken in Australia between 2021 and 2023. In-situ observations of $CH_4$ and $CO_2$ mole fractions were made onboard an aircraft operated by Airborne Research Australia. We assess $CH_4$ and $CO_2$ emissions estimated using a

mass balance method by comparing them to operator-reported $CO_2$ emissions from the same measurement period. We further analyse the use of the co-emitted $CO_2$ tracer correlation method to estimate $CH_4$ emissions from LNG facilities by relying on the reported $CO_2$ emission rates. Section 2 describes the airborne instrumentation, measurements and emission quantification approaches. In Section 3 we present results of the $CO_2$ emission comparison, along with the observed $CH_4:CO_2$ ratios and the $CH_4$ emissions inferred from these ratios. Finally, we

discuss these results and the implications for monitoring $CH_4$ emissions from LNG and other O&G facilities, particularly in the context of measurement-based reporting mechanisms such as OGMP 2.0.

## 2 Methods

### 2.1 Aircraft instrumentation

Atmospheric measurements of $CO_2$ and $CH_4$ were conducted onboard a Diamond aircraft HK36TTC-ECO Dimona

operated by Airborne Research Australia. Further details are given in the SI Text. Dimona aircraft have been used previously for studies into $CH_4$ emissions from coal seam gas production (Neininger et al., 2021), feedlot $CH_4$ and $NH_3$ emissions (Hacker et al., 2016). The scientific instruments were housed in two underwing pods each capable of holding 55 kg and two underwing pylons holding 15 kg. Two GPS-IMU systems (OXTS RT4003, Oxford Technical Solutions, Oxford, UK and Novatel CPT770, Hexagon, Calgary, Canada) recorded high-resolution

aircraft parameters including GPS position, altitude, ground speed, air speed, attitude and accelerations. All aircraft parameters were recorded at 250 Hz (RT4003) and 200 Hz (CPT7700). A high-resolution turbulence probe (BAT probe, Hacker and Crawford, 1999) was used to record air angles and ambient temperature and pressure to yield horizontal and vertical wind components at 20 Hz. Air temperature and dew point were also measured using a Meteolabor TP3 system at 20 Hz. Measurement precisions of these meteorological observations were 0.3 m s$^{-1}$ for

wind speeds, 0.5 K for temperature and 500 Pa for ambient pressure. In addition, a MetOne particle sensor was flown as well as a Riegl LD90 Lidar altimeter.





Atmospheric mole fractions of $CH_4$ and $CO_2$ were measured using an ABB Los Gatos Research Ultraportable Greenhouse Gas Analyzer (UGGA). The manufacturer specifications were a precision of 1.4 ppb for $CH_4$ and 0.3

ppm for $CO_2$ at 1 Hz. The response time was less than 2 seconds with the addition of an external MZ 2C Vario pump with a 2.5 $m^3$ $h^{-1}$ pump speed. The instrument was housed in the underwing pod on the left-hand wing, with a ~15 cm long air intake from the outside. Response time correction was applied based on simultaneous measurements from a Licor LI-7500 open path $CO_2$/$H_2O$ analyser mounted in the same pod. The external pump was mounted in the luggage area of the cockpit.


$CH_4$ and $CO_2$ calibration for the UGGA was applied immediately post-flight through sampling of a secondary standard gas certified by the Commonwealth Scientific and Industrial Research Organisation (CSIRO). A single level reference was used with a target dry mole fraction of 1844.3 ±0.6 ppb for $CH_4$ and 409.90 ±0.01 ppm for $CO_2$, with an assumed linearity in response between 0.01 – 100 ppm for $CH_4$ and 1 – 20000 ppm for $CO_2$ based on

manufacturer specifications. A one-point offset correction was applied to $CH_4$ and $CO_2$ data from each flight based on the mean mole fraction measured for the standard gas.

## 2.2 Flight strategy and mass balance approach

To estimate site-level emissions of $CH_4$ and $CO_2$ from each of the four LNG facilities (which we label A-D), we

followed a single screen mass balance approach (France et al., 2021; Hacker et al., 2016; Krings et al., 2018). Here, the horizontal and vertical variation in atmospheric mole fractions were mapped by measuring along approximately perpendicular flight tracks downwind of each LNG facility. Compared to box patterns, the single screen approach has the advantage of efficiency, since more time is spent measuring the downwind plume, rather than background mole fractions. However, it does rely on relatively strong and stable winds to ensure no mass is lost through the 3

unmeasured sides of a theoretical box surrounding each LNG facility. Real-time $CO_2$ and $CH_4$ mole fraction measurements and meteorological data were used to guide the flight path for measurement optimization, further described in the SI text.

The flight strategy was designed to measure a number of vertically stacked (approximately) constant altitude

transects, with a vertical spacing of approximately 250 ft (76 m). Multiple transects were flown along the same heading, typically from as low as 75 m a.g.l., until no enhancements were detected above the top of the plume, or





the convective boundary layer was reached (as determined by the onboard meteorological observations), whichever was lower. In almost all examples in this work, the vertical extent of the plume was less than the convective boundary layer, and plumes topped out below 1200 m a.g.l. Each collection of vertically stacked transects makes up a curtain and results in one emission rate quantification. As far as possible, curtains were flown perpendicular to the mean wind direction, although some curtains deviated from this by up to 33° (see SI Fig. S1). The number of curtains performed per flight varied from one to six, and each curtain took between 30-90 minutes to complete. A total of 83 curtains were successfully performed across the four sites in this study on 22 days. For sites A-D respectively there were 28, 17, 20 and 18 quantifications. Figure 1 shows the horizontal and vertical distribution of airborne $CH_4$ and $CO_2$ measurements from one of the downwind curtains performed as part of this study to quantify emissions from one of the LNG sites. Across the 83 curtains, the mean number of transects per curtain was 9, with a range of 4-19. The mean number of 1 Hz in-plume $CH_4$ measurements per transect was 49, roughly equivalent to a 2 km wide plume, for an aircraft flying at 40 m s$^{-1}$.



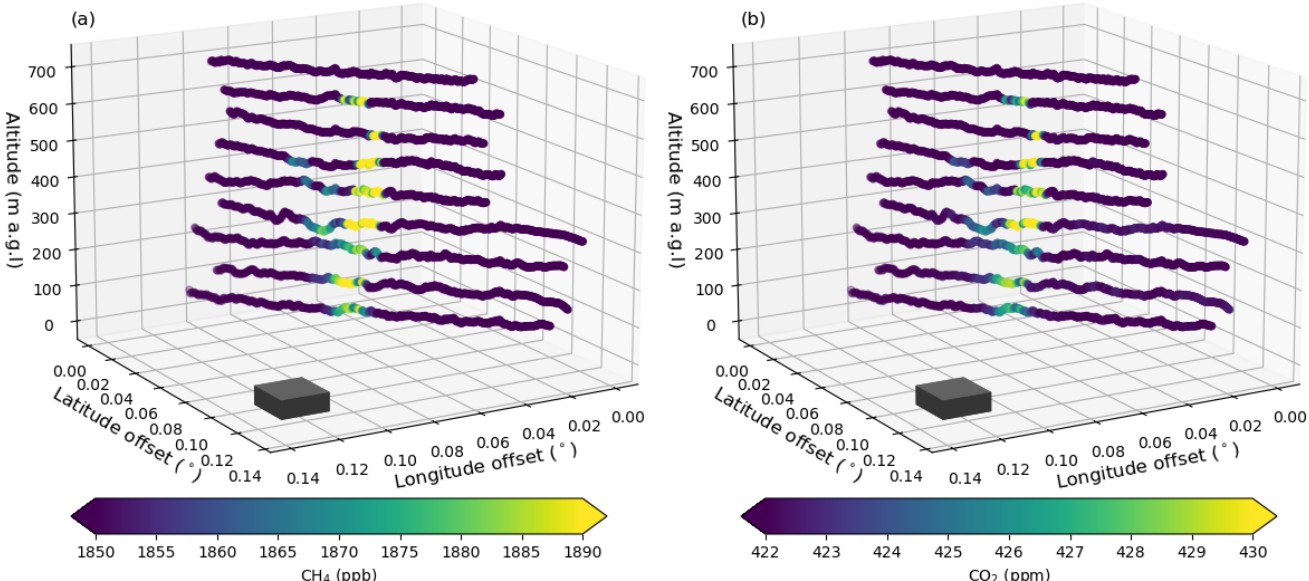

**Figure 1: 3D representation of one of the downwind curtains performed in this study and the corresponding enhancements in (a) CH₄ and (b) CO₂. The grey cube represents one of the LNG site, from which emissions were quantified. Horizontal dimensions are plotted as offsets relative to the most south westerly point. The transects shown were performed 9 km downwind and average 12 km across. The average plume width was approximately 2 km. The colour bar shows the measured mole fractions of CH₄ and CO₂.**

The site-level emission rate, E, was derived from the integral of mass fluxes derived at each sample point in the downwind curtain following Eq. (1):

$$E = \iint \Delta C_i \cdot \frac{\rho_i M}{RT_i} \cdot v_{\perp i} \, dx \, dz \qquad (1)$$

Where $\Delta C_i$ is the dry mole fraction enhancement (CH₄ or CO₂) above background at point i, $\rho_i$ is the ambient pressure, $T_i$ the temperature, M the molar mass of either CH₄ or CO₂, R the molar gas constant, and $v_{\perp i}$ the perpendicular wind speed. The mass flux at each point is integrated across the horizontal (x) and vertical dimensions (z) to give the total emission rate.



To calculate the enhancement above background, $\Delta C_i$, the background itself must first be defined. This was estimated independently for each individual transect and relied on at least 10% of measurements within each

transect being outside the direct outflow of the LNG site emissions and representative of background mole fractions. In practice this was achieved by the pilot/scientist following the signals seen on the onboard computer which gave a live reading of the measured $CH_4$ and $CO_2$ mole fractions. Transect flight tracks were designed to extend until the levels of both $CH_4$ and $CO_2$ had returned to background levels. The transect edges were used for background calculations rather than upwind transects due to greater flight efficiency and closer proximity in time

to the in-plume measurements. For all sites, additional circles around the site were flown to check for the presence of non-LNG upwind sources. For the emission calculations the background was defined as follows:

1. The highest quantile above the first decile within two times the measurement precision of 1.4 ppb for $CH_4$ and 0.3 ppm for $CO_2$ was established.

2. A threshold level was determined as the identified quantile plus two standard deviations of all data less than the quantile. Any data less than the threshold were defined as background. Data above the threshold were temporarily excluded.

3. To account for data noise the background data were smoothed using a 10 s rolling mean.

4. Any gaps in the smoothed background were filled using linear interpolation (i.e. background levels within

plumes were estimated based on an interpolation of the nearest values outside the plume).

5. Any data gaps at transect edges were filled using nearest neighbour extrapolation.

6. The smoothed background field was subtracted from the raw $CH_4$ and $CO_2$ transect measurements to generate the mole fraction enhancements.





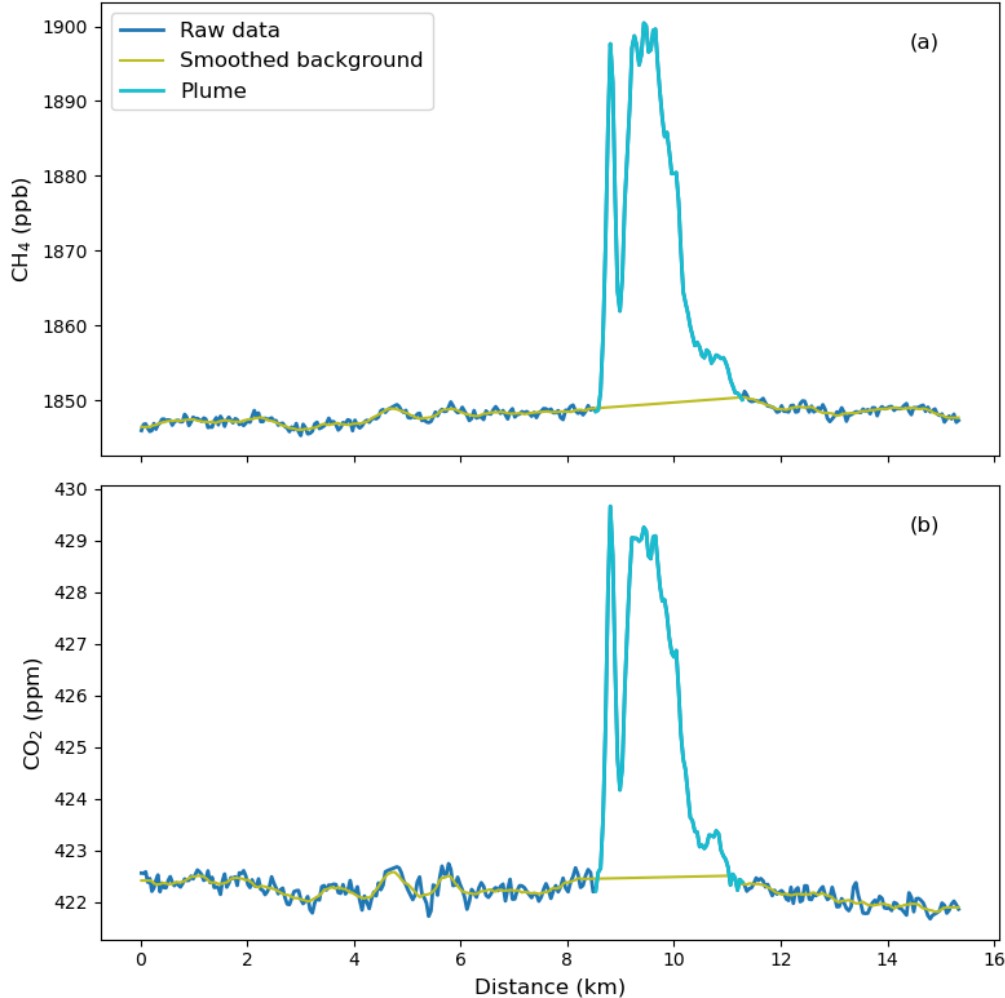

**Figure 2: Example transect data showing the estimated smoothed background mole fractions and the delineation of the downwind plume for (a) CH$_4$ and (b) CO$_2$. The transect was performed at a mean altitude of 150 m a.g.l.**


Figure 2 shows an example of this approach to background definition including the interpolation of values where the data are defined as plume. In the case where downwind transects measured more than one source, the definition

of what mole fractions belonged to the plume of each LNG facility was defined based on visual inspection of the data, combined with a projection of the likely plume origin based on the accompanying measured wind direction and knowledge of the potential sources in the region. Maximum and minimum plume extents were defined for each transect and each site based on latitude and longitude. Only those data within this spatial range were included in the emission calculation for each site. This was particularly critical for two of the sites where a large neighboring

methane source was identified in the data. Due to the prevailing wind direction the emissions plume from the non-





LNG source often overlapped with the emissions plume of the LNG site and the two signals were non-separable. We excluded all instances where the LNG sites could not be confidently separated from other sources from the analysis.

The lowest measurement altitude was typically 75-150 m. Mole fraction enhancement values from the lowest flight level were assumed constant to the surface. Typically, the largest enhancements of either $CH_4$ or $CO_2$ were not found in the lowest transect, minimizing the impact of this assumption. Where transects were conducted above the plume top, the top of the plume was assumed to be halfway between the uppermost layer with zero enhancement, and the highest layer with a detected enhancement. In most curtains, at least one measurement transect was

performed above the top of the plume enhancements. This was possible since the downwind direction was generally over water with a comparatively shallow planetary boundary layer. On occasions where this was not possible, the mole fractions in the top-most transect were extrapolated to the height of the mixing layer as estimated based on the NCEP Global Forecast System (GFS) 0.25° horizontal resolution meteorology. The flux at the top of the mixing layer was assumed to be half the value of the uppermost transect. Mixing layer estimates were generated by running

the Hybrid Single Particle Lagrangian Integrated Transport (HYSPLIT) model at the time flown by the aircraft. The resulting mass flux data from Equation 1 was integrated across the source-attributed transect data, to create a discrete transect flux. Each discrete transect flux was integrated across the vertical domain following a trapezoidal approach to calculate the total emission rate (e.g. Erland et al., 2022; Foulds et al., 2022; Pühl et al., 2024).

The uncertainty in each emission quantification was estimated based on estimates of the uncertainty in the underlying measured variables, variations to the upper and lower extrapolations, as well as an estimate of the uncertainty caused by sampling errors due to the heterogeneity in the vertical distribution of the plume. We included the impact of a ±0.3 m s⁻¹ variation in wind speed, ±500 Pa variation in pressure and ±0.5 K variation in temperature, based on the precision of the aircraft's meteorological instrumentation. We considered the effect of a

±0.3 ppm and ±1.4 ppb uncertainty in the mole fractions of $CO_2$ and $CH_4$ respectively. Uncertainty in the definition of baseline was estimated by varying the background mole fractions by ±1 standard deviation of the background data. The uncertainty in the top-level extrapolation was examined by varying the flux at the flux extrapolation by ±50%. The uncertainty in surface extrapolation was similarly achieved by varying the assumed surface flux by ±50%. To estimate the impact of sampling errors, we performed sequential quantifications of each curtain using

(n-1) transects, removing a different transect each time. The uncertainty due to the sampling was then estimated as the standard deviation of the emissions estimated using (n-1) transects. A well-mixed boundary layer with multiple



transects should have a low standard deviation, and the sampling error would be limited. At the other extreme, quantifications from a poorly mixed layer with few transects that only has a large enhancement in one transect will be heavily impacted by the removal of that transect from the curtain. We assumed the uncertainties of individual
components were independent of each other and summed the relative uncertainties in quadrature to calculate the total uncertainty of each quantification. The relative contribution of these different uncertainties to total uncertainty is discussed in the results section.

## 2.3 $CO_2$ tracer correlation approach

A second approach to estimate $CH_4$ emissions relies on the tracer correlation approach. Here, the ratio of measured $CH_4$ dry mole fractions to mole fractions of another co-tracer gas is used. Typically, the co-tracer is released deliberately from locations co-located with anticipated sources of methane at a known, controlled rate (Roscioli et al., 2015; Yacovitch et al., 2017). Provided emissions occur from the same point (and the same gas temperature), the ratio of mole fractions should be equivalent to the molar ratio of emissions. Even in the case of sources that are
not exactly co-located, this assumption may hold provided the plumes are well mixed i.e. the measurements are taken far enough downwind for the sources to appear co-located. Although the study was not set up with a tracer release in mind, here we utilised the fact that $CO_2$ emissions from LNG facilities should be relatively well-known, since $CO_2$ emissions are predominantly the result of fuel combustion, the consumption of which is required to be quantified in order to meet National Greenhouse and Energy Reporting (NGER) emissions reporting requirements.
The NGER scheme is the Australian national framework for reporting company information about greenhouse gas emissions, energy production and energy consumption. Some sources of $CO_2$ within an LNG facility are also potential sources of $CH_4$, such as gas turbines, flares and vents (Innocenti et al., 2023).

When multiplied by the known emission rate of $CO_2$, $E_{CO2}$, the $CH_4$ emissions rate, $E_{CH4}$, can be estimated following
Eq. (2):

$$E_{CH4} = \frac{[CH_4]M_{CH4}}{[CO_2]M_{CO2}} \cdot E_{CO2} \qquad (2)$$

Where [$CH_4$] and [$CO_2$] are the measured mole fractions and $M_{CH4}$ and $M_{CO2}$ the molecular weights of the respective
gases. The known release rate of $CO_2$ is taken from operator records. The advantage of this method is that it is





unaffected by uncertainties in the wind speed and direction, or assumptions that multiple transects measured over a period of minutes to an hour represent an instantaneous downwind curtain. The disadvantages are that we must assume the emissions of $CO_2$ are sufficiently well known, and that $CH_4$ and $CO_2$ sources are sufficiently co-located, released at the same temperature, sufficiently well mixed at the point of measurement, or an adequate number of

samples are collected so as to be representative at a site level.

To account for the potential differences in source location we explored two ways of calculating the $CH_4$:$CO_2$ ratio. The first approach calculated the ratio using an ordinary least squares (OLS) regression of 1 Hz dry mole fraction enhancements of $CO_2$ and $CH_4$ in each transect plume following Eq. (3).


$$R_{OLS} = \frac{\sum_{i=1}^{n}(X_i - \bar{X})(Y_i - \bar{Y})}{\sum_{i=1}^{n}(X_i - \bar{X})^2} \qquad (3)$$

Where $X_i$ is the magnitude of each 1 Hz $CO_2$ mole fraction enhancement in the plume and $Y_i$ the magnitude of the $CH_4$ enhancement.  Background levels were subtracted from the raw data following the same procedure in Section

2.2, and data outside the plume were ignored. Regressions were only performed where both $CH_4$ and $CO_2$ mole fraction enhancements were greater than two times the measurement precision above background. This means any $CH_4$-only or $CO_2$-only plumes were ignored, which could impact the accuracy of resulting $CH_4$ emission estimates. To address this, we used a second approach, where we calculated the ratio of the plume areas following Eq. (4):

$$R_{Area} = \frac{\int_{plume\ start}^{plume\ end}[CH_4] - CH_{4\ bg}\ dx}{\int_{plume\ start}^{plume\ end}[CO_2] - CO_{2\ bg}\ dx}, \qquad (4)$$

Where $CH_{4\ bg}$ and $CO_{2\ bg}$ the estimated background levels. This may better account for non-co-located sources, which might occur in the case that $CH_4$ is not emitted from the same locations as the combustion dominated $CO_2$ sources. We compare both methods of calculation in Section 3.6.1.


### 2.4 Operator $CO_2$ emission estimation

To assess the quantification performance of the mass balance approach and to use the tracer correlation approach, we rely on operator estimated emission rates of $CO_2$. For two of the sites these were estimated based on the





protocols of the NGER scheme using default factors for flaring and compositions sourced from gas chromatographs
and periodic sampling for Acid Gas Removal, power generation and the LNG train fuel combustion, alongside near
continuous monitoring of gas flow rate through the different elements of the sites. This resulted in estimates of $CO_2$
emissions every 10 minutes. For comparison to the aircraft-based mass balance estimates, data were aggregated
into hourly means and results compared to the nearest hour. On the days of comparison, the average standard
deviation of estimated 10-minute emissions during a 1-hour measurement period was 0.5% of the mean emission
rate, suggesting emissions were essentially constant during the sampling periods. The operator-estimated
uncertainty on the $CO_2$ emission rates was ± 9 and ±12% at the 95% confidence level which we use for the basis
of comparisons and the tracer correlation approach.

For the remaining two sites, $CO_2$ estimates also rely on methods used to estimate emissions following protocols in
the NGER scheme. Emissions were provided as monthly totals, consistent with the internal approach to quality
assurance. Calculation of emissions from turbines was made using NGERs emissions factors considering measured
fuel calorific value, incorporating direct measurement of key variables such as fuel gas mass flow rate to turbines,
at short time steps, and incorporating periodic composition analyses. Based on operating conditions, and default
NGER uncertainties, aggregated uncertainties in these estimates are estimated to be ± 20% and ± 25% respectively
at the 95% confidence level. We use this as the 2σ uncertainty for the basis of comparisons. These uncertainties are
larger than the other sites due to the estimation methods and the different contributions of combustion sources
versus other sources of $CO_2$. One site underwent planned maintenance to an LNG train during the period of
observations. There was a 4-week gap between the first and last day of measurements at this site and operating
conditions changed during this period (e.g. start-up flaring, gas turbine throughput). Although the aggregated total
monthly emissions estimate is unlikely to be equivalent to individual daily or hourly emissions values, we still
include this site in some of our comparisons.





**3 Results**

**3.1 Mass balance CO₂ emissions evaluation**

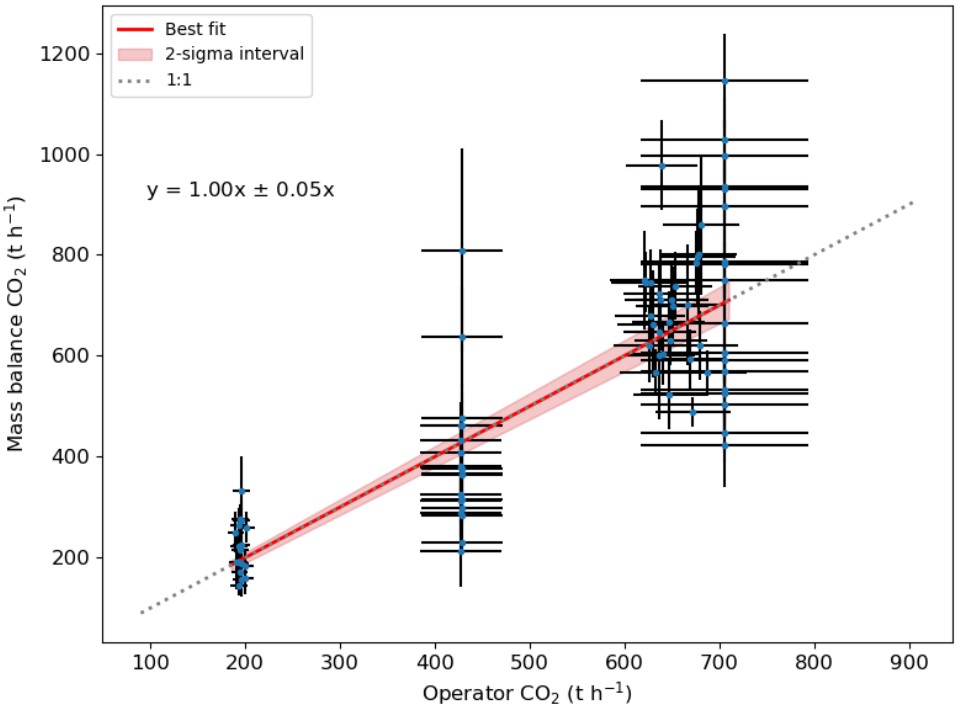

**Figure 3: Comparison of operator CO₂ emission rates with airborne mass balance CO₂ estimates in t h⁻¹. Uncertainty bars represent**
**2σ level.**

Figure 3 shows the comparison of individual mass balance quantifications of $CO_2$ emissions compared to operator estimates from each of the four sites. The site-level emission rates cover a range of approximately 200 – 700 t h⁻¹, and the number of sampling days at each site ranged from 5 – 10. The best fit is calculated using orthogonal distance

regression to account for errors in both sets of estimates and has a slope of $1.00 \pm 0.05$ (2σ). This indicates there is no significant bias between the mass balance and operator estimates of $CO_2$ emissions across the 4 sites. The mean relative difference of individual quantifications is ±20%. However, this is affected by the comparison to the monthly aggregated total emissions at two of the sites. As noted in Section 2.4, operating conditions are known to have varied throughout the measurement period at one of these. On an individual site basis, the mean relative

differences of individual quantifications are 13%, 22%, 25% and 26% for sites A-D respectively. For sites A and B with high-temporal resolution operator estimates, 43 out of 45 airborne quantifications fall within ±50% of the



operator value at the time of measurement. For sites C and D, 36 of 38 airborne estimates are within ±50% of the monthly aggregated total operator emissions estimate.

### 3.2 Impact of sampling variables on $CO_2$ mass balance estimates

We examined how the absolute difference between operator emission rates and the mass balance estimates for $CO_2$ varied as a function of the downwind distance of the curtain and wind speed. Results from Site D were omitted due to the known dependence on the changing source emissions rate. Across Sites A-C, we did not find any strong correlation between the differences of individual quantifications and the downwind sampling distance (Pearson correlation coefficient of -0.1), or wind speed (r= -0.4). This may be because the range of errors was relatively small and there were not enough data with large errors to form robust correlations. However, when aggregated into bins based on wind speed the impact is more apparent. This is shown in Fig. 4. The mean relative difference to operator reporting for quantifications with a mean wind speed of 3-6 m s$^{-1}$ was 30% (n=16), compared to 18% (n=24) for a 6-10 m s$^{-1}$ wind speed and 13% (n=25) for 10-14 m s$^{-1}$. The wind speed can explain some of the differences between quantifications at different sites, with sites A and B having the largest mean wind speeds and the smallest relative differences.





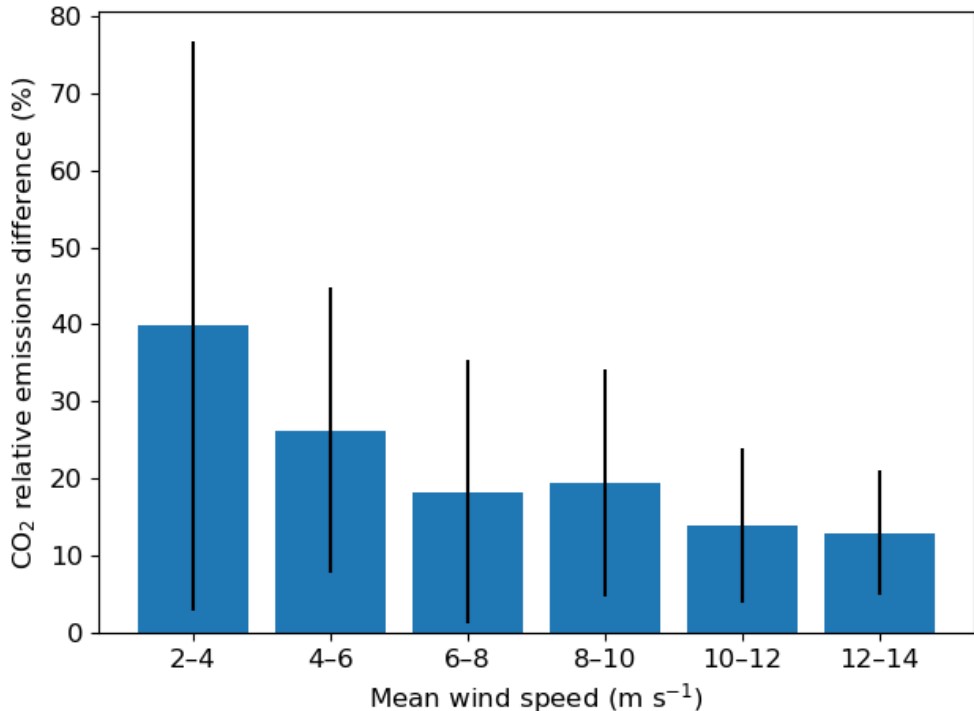

**Figure 4: Impact of wind speed on the difference between mass balance and operator $CO_2$ emission estimates at Sites A-C. The plot shows the mean of individual quantification differences grouped into 2 m s$^{-1}$ wind speed bins. The number of quantifications in each bin from lowest wind speed to highest were: 4, 12, 11, 13, 9 and 16. The error bars show the standard deviation of differences within each group.**

The impact of the number of quantifications on the accuracy of the mean $CO_2$ mass balance estimate is reflected in the 2σ standard error under different wind speeds. At wind speeds greater than 10 m s$^{-1}$, the 2σ standard error of 3 independent samples is 15%, indicating a high level of certainty of average emissions with relatively few quantifications. For wind speeds between 3-6 m s$^{-1}$, the same level of certainty would require around 15 quantifications, although even 3 independent estimates would constrain emissions to within 35% uncertainty at these lower wind speeds. We note that there were additional flights or curtains flown where the average wind speed was below 3 m s$^{-1}$ and the wind direction was variable and unsuitable for a mass balance calculation. An example is shown in the SI, Fig. S2, which was excluded from our analysis. This shows the potential for much larger errors in the mass balance estimates when the assumptions of constant wind speed and direction do not apply.




### 3.3 Assessment of estimated individual quantification uncertainties

Individual quantification uncertainties for $CO_2$ and $CH_4$ were estimated as described in Section 2.2. Across all sites the mean estimated uncertainty at the 95% confidence level was 27% for $CO_2$ and 32% for $CH_4$. The largest component of these uncertainties was the sampling uncertainty, with a mean $2\sigma$ uncertainty of 18 ±10% for the

$CO_2$ estimates. The sampling uncertainty effectively provides a measure of the uniformity of mass flux throughout the plume. Curtains with fewer transects and large variability in mass fluxes between transects have a larger uncertainty than curtains with a higher transect density and more uniform distributions. The contribution of uncertainties due to background definition, $CO_2$ measurement precision and wind precision increased as the emissions strength decreased and wind speed decreased, reflecting the smaller signal-to-noise ratio. However, the

mean $2\sigma$ uncertainty was below 8-9% for all three. Other uncertainties such as surface and boundary layer extrapolation were small, averaging 4-8% $2\sigma$ uncertainty. The largest enhancements in the $CO_2$ and $CH_4$ plumes were generally lofted above the lowest altitude flown, limiting the impact of the surface extrapolation. In addition, measurements for most curtains were conducted above the top of the downwind plume, minimizing uncertainties associated with top level extrapolation.


The typical transect density for each curtain was a regular spacing of 76 m (250 ft), with a mean of 9 transects per curtain. To determine the impact of transect density on the $CO_2$ emission estimates we ran a sensitivity test using only a maximum of half the number of transects, with 150 m average spacing. We removed every other transect, beginning with the lowest transect from each curtain. Removing the lowest and often the highest transect should

increase the impact of assumptions related to the surface and boundary layer extrapolations. However, the impact of the reduced sampling density on the emissions estimates was relatively limited (see SI Fig. S3). Across all quantifications, the mean difference between mass balance and operator $CO_2$ estimates increased from 13 t h$^{-1}$ (3%) to 18 t h$^{-1}$ (5%). The mean absolute error (MAE) increased from ±105 t h$^{-1}$ (20%) to ±120 t h$^{-1}$ (23%). The increase in MAE occurred predominantly for curtains with higher wind speeds, with the absolute error essentially unchanged

for curtains with a wind speed less than 6 m s$^{-1}$. The results of the sensitivity test show that the density of sampling has only a small impact on the derived results, and the accuracy on individual quantifications is unlikely to be improved greatly by sampling densities greater than every 75 m, for the 3 – 15 km downwind distances of the curtains. Whilst sparser sampling leads to slightly larger errors and variation of estimates, 88% of the $CO_2$ mass balance estimates remained within ±50% of the operator estimates.





## 3.4 Applicability of $CO_2$ mass balance verification to $CH_4$ mass balance estimates

The comparison to operator data showed there is no significant bias in the mass balance $CO_2$ estimates relative to the operator estimates. In addition, the mean absolute percentage error of individual quantifications was 20%. Given the only difference between $CO_2$ and $CH_4$ quantifications is the mole fraction enhancement, one might expect a

similar quantification performance for $CH_4$. If both $CO_2$ and $CH_4$ emissions are relatively constant, and the random uncertainties are similar, then this should be reflected in the variance of the $CO_2$ and $CH_4$ estimates. From our observation-based emission estimates, it is apparent that $CH_4$ emissions often varied on a day-to-day basis. However, estimates from the same day tended to cluster around a similar emissions rate. Therefore, we compared the magnitude of $CH_4$ and $CO_2$ estimation variance by comparing the anomalies of individual quantifications

relative to the daily mean estimate. Any days with only one quantification were omitted from this analysis.

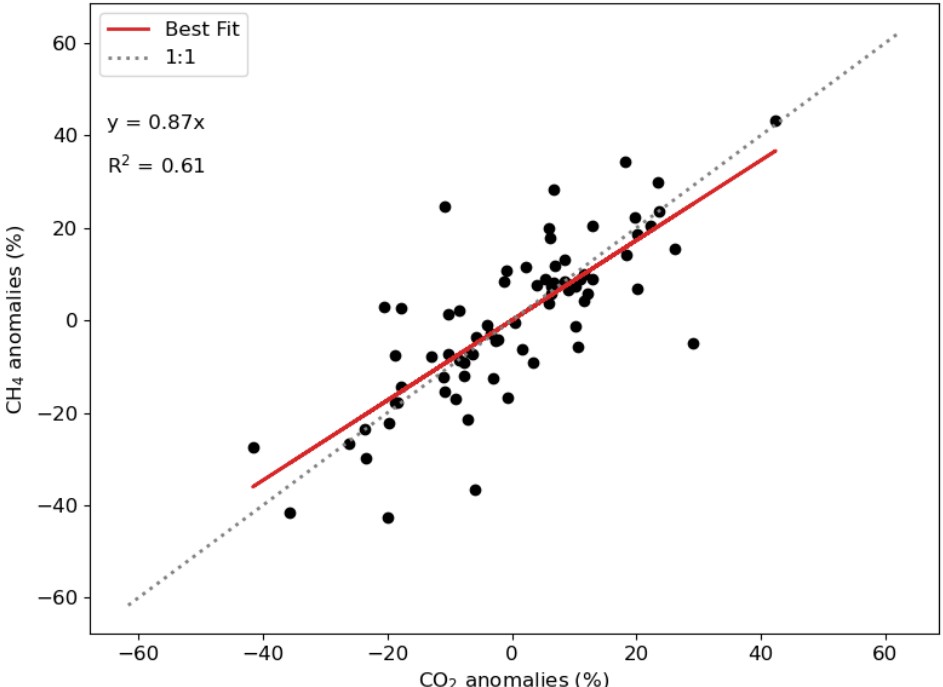

**Figure 5: Comparison of $CO_2$ and $CH_4$ mass balance anomalies relative to the daily mean. The red line shows the ordinary least squares fit.**

Figure 5 shows this comparison of anomalies in $CO_2$ and $CH_4$ mass balance estimates. The plot shows a positive

correlation between $CO_2$ and $CH_4$ anomalies, with an ordinary least squares regression gradient of 0.87, indicating similar random variation of the $CO_2$ and $CH_4$ mass balance estimates. The daily standard deviation of 10-minute operator $CO_2$ estimates was 3% on average across the days of measurement. This suggests that most of the variation





in the daily $CO_2$ estimate anomalies was due to random quantification error, rather than emission variation. Given that similar relative magnitudes are apparent in the $CH_4$ anomalies, this implies similar random quantification
uncertainties apply to $CH_4$.

The resemblance of $CO_2$ and $CH_4$ random errors can be understood in terms of the common features and assumptions in the method such as wind speed, pressure, interpolation and sampling density. Furthermore, the similar size of measured enhancements above background relative to the instrument precision might explain the
similar performance. The mean enhancement in a downwind plume at 4 km distance with a 10 m s$^{-1}$ wind speed was 13 ppm for $CO_2$ and 50 ppb for $CH_4$. Compared to the stated instrument precision of 0.3 ppm and 1.4 ppb for $CO_2$ and $CH_4$ respectively, these are both around 35-40 times larger. Based on this, and the comparison shown in Figure 4, it is likely that the $CH_4$ emissions quantification accuracy will follow the performance characteristics found for $CO_2$.


The intercept of the best fit line in Figure 5 is inevitably zero because the mean of each set of daily anomalies is zero. The comparison of random errors does not address the potential for bias in the $CH_4$ estimates. However, since the $CO_2$ estimates were shown to be unbiased relative to operator reported emissions, and the random errors between $CH_4$ and $CO_2$ quantifications are similar and rely on many of the same variables, it is likely that the mass
balance $CH_4$ estimates are also unbiased relative to the true emissions, unless systematic errors are introduced via the $CH_4$ mole fraction measurements, or sampling of the vertical distribution of the $CH_4$ plume.

### 3.6 Measured $CH_4$:$CO_2$ ratios

In this section, we examine measurements of the $CH_4$:$CO_2$ dry mole fraction enhancement ratio, and the resulting
$CH_4$ emission estimates. Figure 6 shows an example of $CH_4$ and $CO_2$ mole fraction enhancements measured on four different days over a 2-week period at one of the sites. The magnitude of the mole fraction enhancements varies depending on factors such as wind speed, downwind sampling distance and boundary layer dynamics. Although the sampling conditions were different on each day, the OLS regression on each day yields relatively consistent estimates of the mole fraction ratio, all within 10% of each other, suggesting the site-level emission ratio
was similarly consistent. The site-level coefficients of determination between $CO_2$ and $CH_4$ enhancements are not as uniform, with the best fit line in Fig. 6 (b) in particular having a poor visual fit. The data on this day were collected at a mean downwind distance of 2.7 km and a wind speed of 4.4 m s$^{-1}$. This represents the closest mean



sampling distance of any day and may explain the poorer site level correlation. As shown by the colour scale in Fig. 6, which denotes the altitude of sampling, different gradients may be inferred at different altitudes (i.e. along

different transects). This is further shown in the SI, Fig. S4-S5, which indicate that the weaker site-level correlations are underpinned by stronger coefficients of determination ($R^2$ >0.8) along the majority of individual transects. This suggests that the ratios of individual transects may be representative of different source emission ratios, or different exit velocities within the LNG site, rather than site-level emission ratios. In addition, poorer correlations at closer distances are likely a result of the inexact co-location of $CH_4$ and $CO_2$ sources, different release temperatures or

exit velocities. In the cases shown here in Fig. 6, the differences between individual transect ratios average out to result in similar site-level emission ratios across the 4 different days, although this may not be guaranteed. The days with poorer overall correlation between $CH_4$ and $CO_2$ enhancements have larger uncertainties in the site-level ratio estimated by the OLS regression. Even so, given the minimum number of in-plume measurements was 436, the maximum uncertainty on the best-fit gradients is ±6% at the 95% confidence level for the four cases shown in

Figure 6.



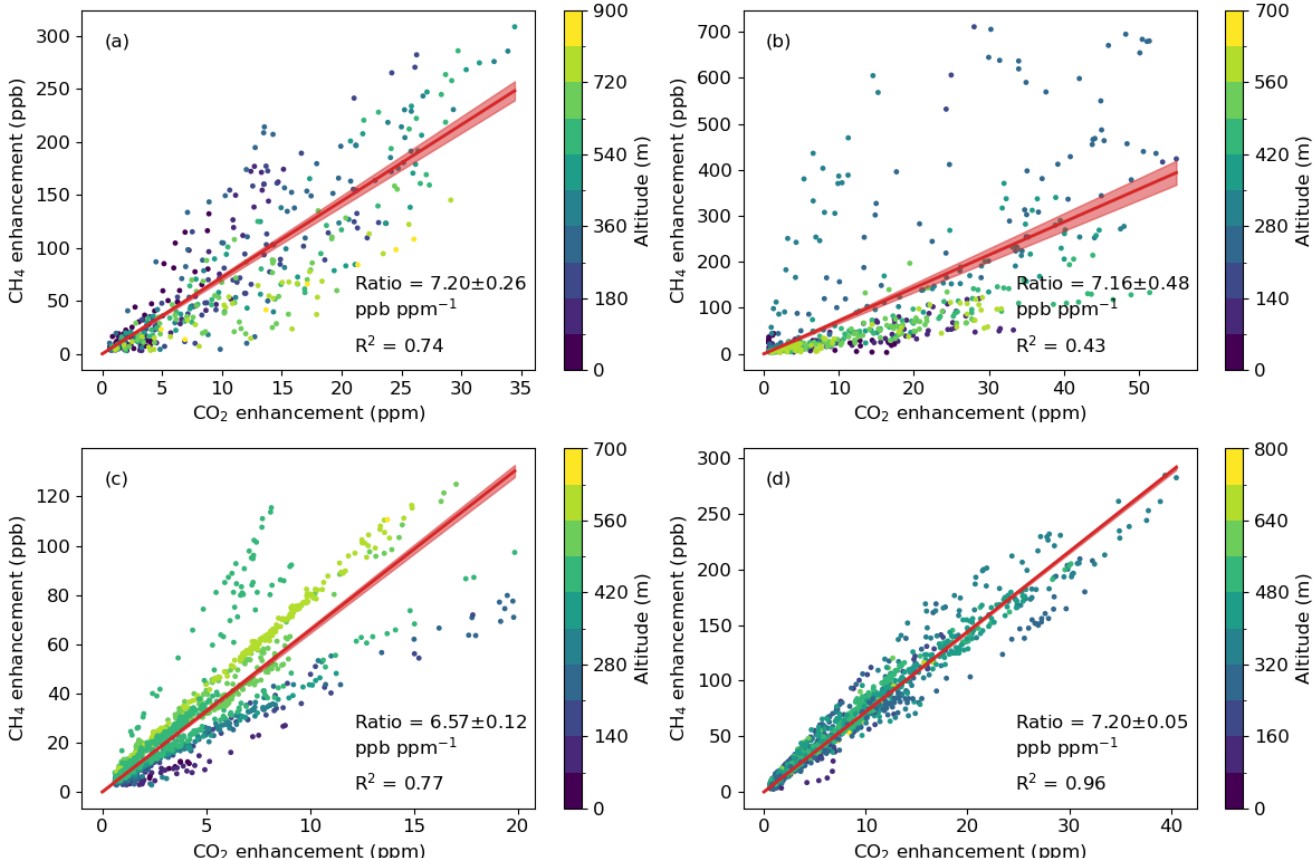

**Figure 6: Scatter plots of $CO_2$ versus $CH_4$ enhancements for one of the sites on 4 different days (a-d). Although the strength of the correlation varies, the ratios calculated using OLS regression are relatively consistent as indicated by the red lines and $2\sigma$ shading. Measurements were performed at mean downwind distances of: (a) 3.0 km, (b) 2.7 km, (c) 3.1 km and (d) 8.0 km. The colour scale indicates the altitude of each measurement point to help show the underlying transect correlations.**

The measurements shown in Fig. 6(c) were collected on a day where the atmospheric conditions precluded the use of a mass balance approach (see SI Fig. S2). Whilst conditions were unsuitable for a mass balance estimate, because the atmospheric $CO_2$ and $CH_4$ were transported under the same conditions, the resulting ratios of the gases shown in Fig. 6(c) still closely resemble the ratios measured on other days. A number of distinct gradients are visible in Fig. 6(c) at different altitudes, possibly indicative of different sources from within the site. However, the overall regression gradient of 6.56 ppb $CH_4$ ppm$^{-1}$ $CO_2$ is within 10% of the mean $CH_4$:$CO_2$ measurement ratio across all days. Even allowing for uncertainty in our knowledge of the true $CO_2$ emission rate, this shows the potential value of the co-emitted $CO_2$ tracer correlation approach for estimating $CH_4$ emissions under conditions where mass balance assumptions fail. Figure 6 (d) shows a case where measurements were performed 8 km downwind, and the





coefficient of determination between $CO_2$ and $CH_4$ enhancements is 0.96. In this case, the ratios measured along individual transects are likely to be more representative of the site-level emissions ratio.

### 3.6.1 OLS versus plume area ratios

Figure 7(a) shows a comparison of individual transect ratios calculated by the OLS and plume area methods described in Section 2.3 and Eq 3-4. Although there are occasional differences, most calculations fall along or close to the 1:1 line, with a mean gradient of 1.01, and $R^2$ value of 0.97. Therefore, the choice of approach to calculate the ratios for individual transects does not have a major impact on the derived values and the resulting $CH_4$ emissions calculation for the cases explored here, since the correlations between $CO_2$ and $CH_4$ enhancements along individual transects are generally strong (SI Fig. S4). However, depending on co-location of emissions, this may not be generalizable to all LNG facilities or if measurements are performed at closer distances. Since the OLS approach has the advantage that it is easier to apply across multiple transects or random samples of the data, we use this approach for all further calculations.





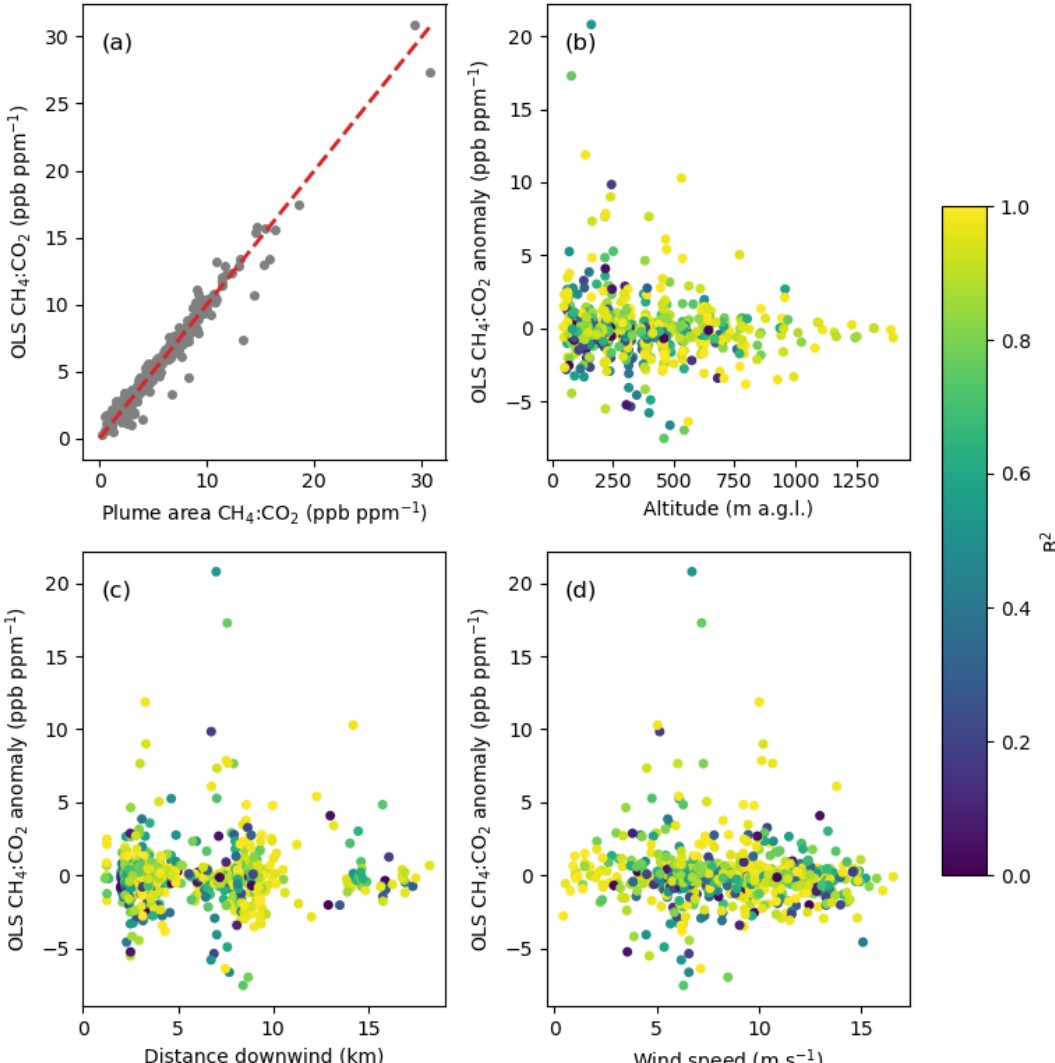

**Figure 7: (a) Comparison of CH₄:CO₂ ratios calculated by OLS and using the ratio of plume areas across each transect. (b) The**
525 **relationship between individual transect ratios and the sampling altitude. (c) Individual transect ratios versus downwind sampling distance. (d) Individual transect ratios versus mean measured wind speed. The colour scale in (b)-(d) represents the coefficient of determination between CO₂ and CH₄ enhancements in each transect.**

### 3.6.2 Impact of sampling variables on measured CH₄:CO₂ ratios

Figure 7(b) shows the impact of sampling altitude on the calculated transect ratio across all sites and dates. To enable sites to be compared to each other and to allow for day-to-day CH₄ emissions variation the data are plotted as relative to the daily mean ratio across all transects at each site. There is no clear trend of increasing or decreasing



ratios with altitude. However, larger variability in positive and negative anomalies are seen at lower altitudes. In particular, large positive anomalies (higher $CH_4:CO_2$) ratios are observed at samples below 250 m. The standard deviation of anomalies below 250 m is 2.75 ppb ppm$^{-1}$, compared to 1.75 ppb ppm$^{-1}$ above 250 m, an increase of 57%. Whilst this may be indicative of poor mixing below this height, the much larger ratios could also be the result of actual emissions variation on that day, especially if the $CH_4$ was emitted from sources with relatively less lofting from buoyancy or heat effects compared to $CO_2$ sources. Since the measurements were often taken over water, other considerations could be the presence of LNG tankers or other ships that might have caused these larger variations. Above 1000 m the anomalies are more tightly constrained around 0, indicating signals more representative of the daily mean ratio. This is consistent with greater mixing before plumes reached the higher altitudes as well as more convective conditions that enable plumes to reach these altitudes. However, the number of transects with detectable plumes above this height was small and they are largely from the same day. Figure 7 (c) and (d) show the relationship between the calculated $CH_4:CO_2$ ratio anomalies and distance or wind speed. Neither variable shows a clear impact on the derived ratios or variability of the ratios, suggesting that wind speed and distance do not have a significant influence on the calculated ratios. Similarly, we found no obvious relationship between the transect $CH_4:CO_2$ ratios and the observed in-situ temperature, (see SI Fig. S6), although it is likely that different exhaust temperatures and exit velocities at the emission sources would impact the vertical distribution of the $CH_4:CO_2$ ratios.





### 3.6.3 Single height ratios vs multi-height ratios

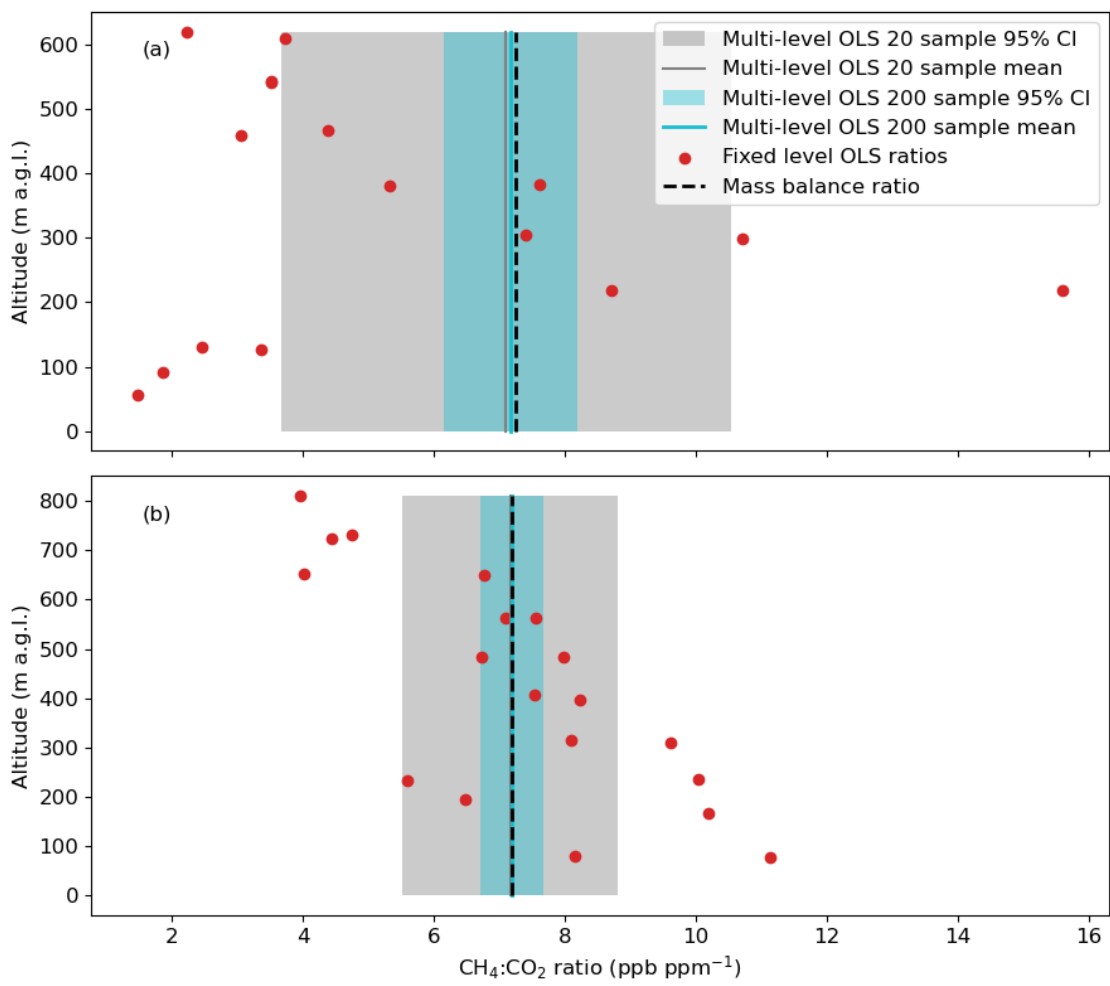

**Figure 8: Individual transect ratios (red) and multi-height OLS regression ratios using 20 (grey) or 200 (cyan) samples. The ratio of the integrated fluxes calculated by the mass balance is also shown (black dash). The shading represents the 2σ uncertainty on the multi-level ratio estimates (CI = confidence interval). The two panels represent different measurement days at the same site, showing how vertical profiles of the ratios may vary day-to-day.**

The comparisons shown in Fig. 7 represent ratios calculated for individual transects, performed at near-constant altitude downwind of the LNG sites, and suggest larger variability of mole fraction ratios at lower altitudes. The individual transect ratios for two particular days at one of the sites are shown in Fig. 8. They show how the vertical profile of the $CH_4$:$CO_2$ ratios may vary on a daily basis, and therefore be difficult to predict. We assume that the ratio of the integrated fluxes of $CH_4$ and $CO_2$, calculated via the mass balance approach provides the best



representation of the true emissions ratio, represented by the black vertical dashed line. On the first day, the measured ratios below 200 m are all at a factor of 2-4 smaller than the ratio of the integrated plumes. However, the second day shows a linear decrease of the mole fraction ratio with altitude. If measurements are only performed at a single altitude on a given day (e.g. lower than 200 m), even if repeated many times, the resulting emissions estimate for that day would be biased low in the first case, and biased high in the second. It is worth noting that the ratio of integrated fluxes remained unchanged between the two days. Therefore, the differing vertical profiles of the measured ratios are more likely determined by atmospheric dynamics than source level changes in emissions. To return an estimate close to the integrated mass balance ratio, a full set of transects would need to be conducted.

However, by conducting random sampling across the full vertical extent of the plume we can retrieve ratios that are much closer to the fully integrated mass balance ratio with far fewer data points. To calculate this, we randomly sample from all data within the downwind plume. In total, there were 756 individual 1 Hz measurements within the downwind plume on the first day and 628 on the second. As shown in Fig. 8, an OLS regression performed on a random sample of 20 individual measurements, returns a ratio that is in the range 3.2 – 10.2 ppb ppm$^{-1}$ at the 95% confidence interval. This is equivalent to 52% uncertainty at the 95% confidence level from 20 samples. On the second day, where there was less variability in ratios across different altitudes, this 95% confidence level uncertainty was 4.8 – 9.5 ppb ppm$^{-1}$, a 33% uncertainty. Figure 8 shows that this uncertainty can be significantly reduced by incorporating further samples into the OLS regression. A 200-sample regression gives $2\sigma$ uncertainties of ±19% and ±8% on the two days. The results show the importance of sampling across the full vertical extent of the plume to retrieve an unbiased estimate of the daily emissions rate using the co-emitted $CO_2$ tracer correlation approach.





### 3.6.4 Comparison of CH₄ emission estimates

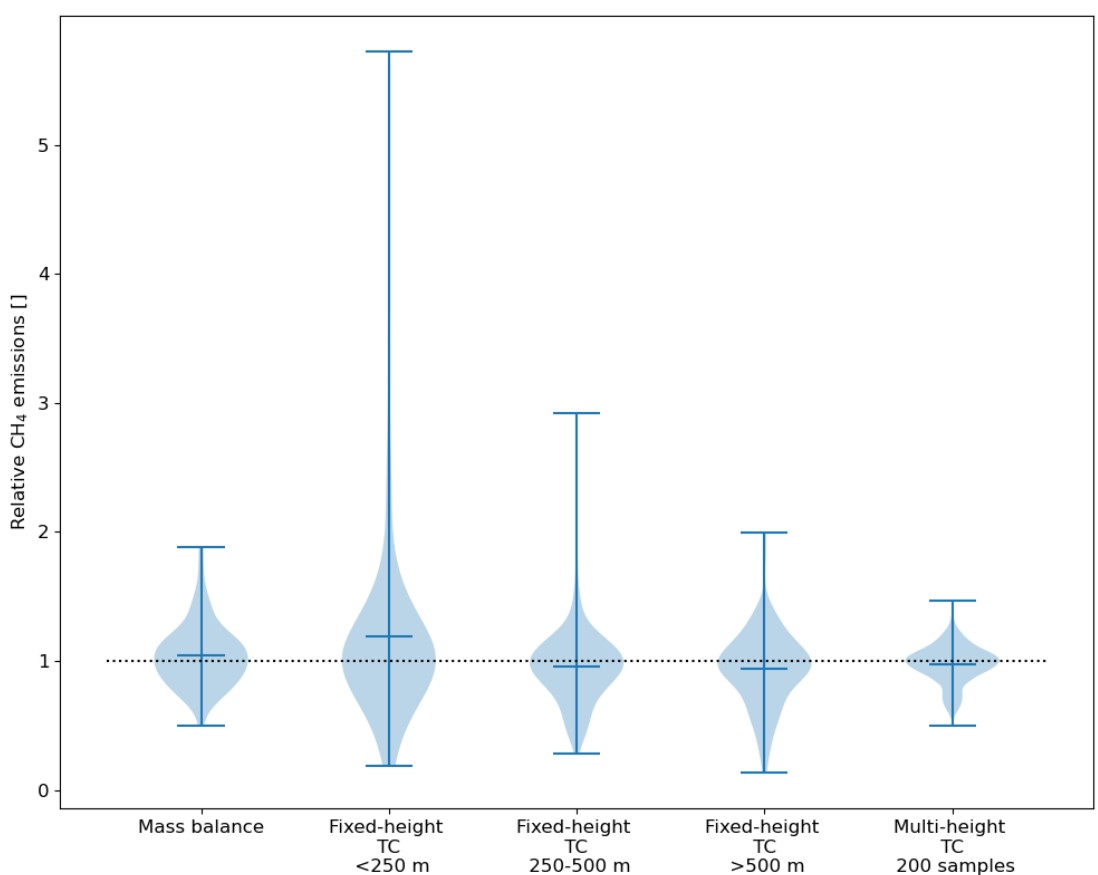


**Figure 9: Comparison of CH₄ emission estimates across dates and sites A-C. Emission distributions are normalized using the daily mean ratio of CH₄:CO₂ mass balance estimates multiplied by the operator CO₂ emission rate . The violins represent the distribution of individual CH₄ emission quantifications using each approach. The first violin shows the distribution of mass balance CH₄ estimates. The final 4 violins show CH₄ emissions estimated based on different implementations of the tracer correlation (TC)**

**approach.**

The measured ratios can be combined with the operator reported $CO_2$ emission rates to estimate $CH_4$ emissions. For sites A and B we use the emission rate at the nearest 10 minute emission time. For site C we use the monthly average emission rate. We ignore site D due to the known changes in $CO_2$ emissions not being accounted for by the monthly aggregated total emissions estimate. The emission time was calculated by estimating the time taken

for the plume to reach the measurement point using the measured wind speed and distance from the site, and subtracting this from the measurement time.





Figure 9 shows a comparison of $CH_4$ emissions estimated via different methods. To ensure more statistically robust distributions, the data from each site are combined by normalizing using the mean ratio of $CH_4$:$CO_2$ mass balance

estimates on each day of sampling multiplied by the operator $CO_2$ emission rate. Provided the operator $CO_2$ estimates are representative of the true emission rate, using the ratio of mass balance $CH_4$ and $CO_2$ calculations to estimate $CH_4$ emissions should be the most accurate approach. This is because it incorporates the full 2D plume integration, and errors in transport assumptions cancel between the mass balance estimates. The violins represent the distribution of individual quantifications from each approach, including those estimated via mass balance. The

mass balance mean is 1.04, since the $CH_4$ emissions estimated via mass balance are slightly larger than those estimated using the ratio of mass balance $CH_4$:$CO_2$ estimates. It is worth noting that the mean mass balance $CO_2$ estimate for sites A-C is 3% larger than the operator value, which largely explains this difference.

The fixed-height ratio estimates represent the distribution of individual transect ratio-based estimates on each day,

divided into altitude ranges of 0-250 m, 250-500 m and 500+ m. There are 172, 170 and 112 members of each of the respective groups. The 200-member multi-height distributions are based on a Monte Carlo sampling approach, with 200 different random sets of 200 observations used on each date to build up the distribution. For the ratio calculations, for each member of the distribution, uncertainty in the operator $CO_2$ emissions was incorporated by sampling from a normally distributed random sample with the mean emissions representing the operator mean, and

standard deviations based on the operator estimated uncertainties.

All estimates across each of the sites overlap within the bounds of their distributions, and the mean of the distributions are all within 1-18% of the mass balance mean. The medians are all within 5% of each other, showing how the ratio-based emissions are similar to those estimated through mass balance. The largest difference is for

emissions estimated using transect ratios below 250 m, where some samples show ratios up to 5 times larger than the mass balance ratio estimate, resulting in a mean estimate 18% larger.

As noted in Section 3.6.3, measurements from a single downwind pass, or even multiple passes on the same day at these altitudes may capture the emission ratio of only part of the site and lead to inaccurate estimates of the overall

site-level emission rate. For estimating a snapshot emission rate, there is a reasonable probability of inaccurate estimate when measuring only below 250 m. However, there is no clear bias in multi-day sampling at these altitudes. Given enough measurements across multiple days, it is likely that the errors may average out, and indeed the median of the distribution is within 3% of the mean mass balance estimate. If conducted over multiple days and





different atmospheric conditions, measurements below 250 m may be suitable to estimate average site level
emissions.

For ratios measured above 250 m and 500 m, the likelihood of extreme values is reduced, but the standard deviations of the distributions are slightly larger than the mass balance deviation at 30%, similarly highlighting the need for repeat samples across multiple days where the vertical distributions differ. For the multi-height ratio, the standard
deviation is 15%, smaller than the variability in the mass balance estimates. We note that a large part of this $CH_4$ emissions uncertainty is caused by the uncertainty in the $CO_2$ emissions which ranged between 9-20% at the 95% confidence level. The distributions show that for individual quantifications, the multi-height ratio method provides the lowest uncertainty estimate. However, all approaches return similar multi-day average estimates with no significant biases.


In the above examples, the ratios were calculated based on the mole fraction enhancements above background levels which first had to be determined following the process outlined in Section 2.2. Through running the same calculations using the raw mole fractions, instead of the mole fraction enhancements, we find that a similar $CH_4$ emissions distribution would be estimated, albeit with larger uncertainties (see SI Text and Fig. S7).






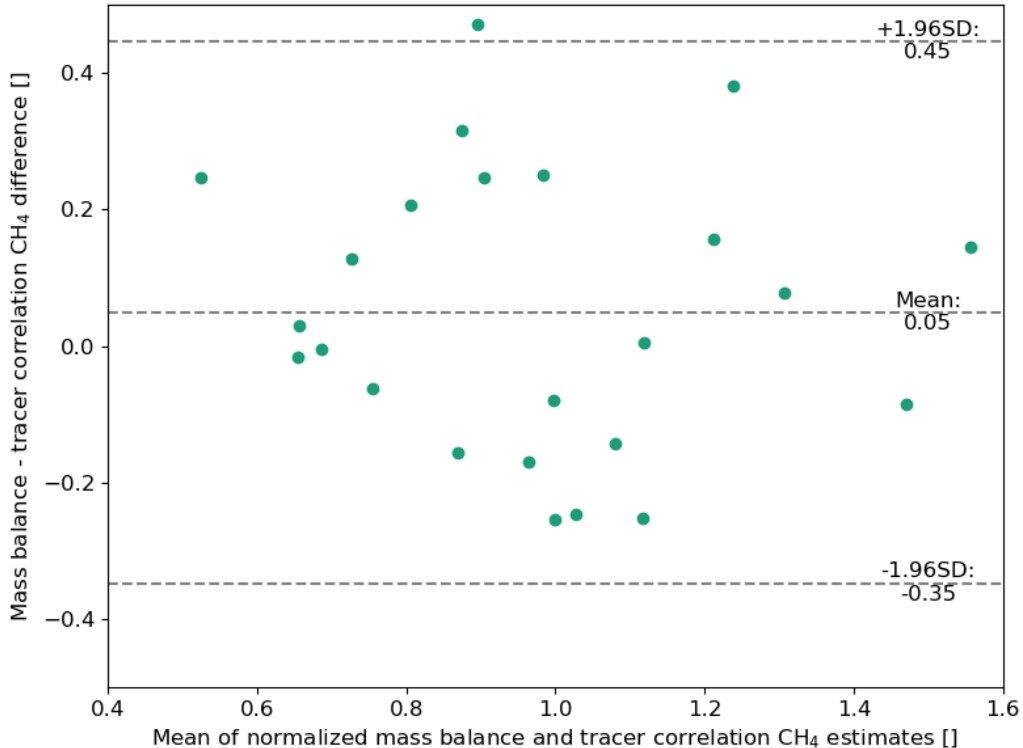

**Figure 10: Bland-Altman plot of the differences between daily mean CH₄ estimates from mass balance and tracer correlation methods. CH₄ emission estimates are normalized by the mean emissions from each site.**

Figure 9 normalizes the CH₄ emission estimates by the daily emission rate, and thus largely removes the temporal variability of emissions and magnitude of emissions from the comparison of distributions. To compare the difference between mass balance and tracer correlation estimates across sites we can instead normalize by the overall mean emissions estimate for each site. Figure 10 shows a Bland-Altman plot comparing the daily mean emission estimates estimated via mass balance and multi-height tracer correlation method using 200 samples. The

plot shows the daily mean CH₄ mass balance estimates are on average 5% larger than the tracer correlation. As noted above, this can largely be explained by the mean $CO_2$ mass balance estimate being 3% larger than operator estimates for sites A-C. 95% of estimates fall within ±40% of each other and there is no indication of a proportional estimation bias at higher or lower emission values. Notwithstanding the 5% offset, Fig. 10 indicates similar distributions of daily mean emissions are estimated by both mass balance and tracer correlation methods.



## 4 Discussion and Conclusions

The results in this study follow a sequential flow. First, the aerial mass balance $CO_2$ estimates were evaluated using operator data. The comparison of $CO_2$ emission estimates using the mass balance method demonstrated that 95% of estimates (79 of 83) fell within ±50% of the operator estimates. This includes the impact of temporal variability for sites where only a monthly operator emissions estimate was available. These results demonstrate the strength of the airborne mass balance approach for quantifying $CO_2$ emissions from LNG facilities. Second, the correlation between errors in $CO_2$ and variability in $CH_4$ estimates suggests similar performance statistics can be expected for $CH_4$ estimates. The approach of measuring and quantifying both $CO_2$ and $CH_4$ allows the $CO_2$ estimates to be evaluated on the fly, under the same conditions under which $CH_4$ emissions (which are more likely to be variable and unknown) are being assessed.

Nevertheless, airborne measurements are often cited as expensive and replicating the mass balance approach applied in this study may not be feasible in other cases. Whilst we note that the cost per flight with the Dimona aircraft was less than USD 10,000, the use of the tracer correlation approach as described here offers the potential for a more efficient and economical means of determining emissions from LNG facilities. This is especially the case where the $CO_2$ co-tracer is already emitted from the target sites and no additional release equipment and procedure is required, what we term a tracer of opportunity. Our results demonstrate that the tracer correlation method, using $CO_2$ as the co-tracer, results in mean $CH_4$ emission estimates that are consistent with mass balance estimates, with a mean difference of 5%. Provided the $CO_2$ emissions are known to a reasonably accurate level the resulting $CH_4$ emission uncertainties may be smaller than the mass balance method, with as few as 200 in-plume measurements.

$CH_4$:$CO_2$ ratios were found to be highly variable across measured transect altitudes on individual days. Measuring at a single altitude (or limited range of altitudes) is likely to lead to unrepresentative site-level emission estimates, with ratios potentially varying by a factor of 10 across altitudes. To attain unbiased, low uncertainty emission estimates for a single monitoring event requires sampling across multiple altitudes, covering the full vertical and horizontal extent of the plume. Here, using only 200 samples, the resulting $CH_4$ emission estimates are likely to be consistent with mass balance estimates, and the uncertainty dominated by uncertainty in the underlying $CO_2$ emissions estimate used to estimate $CH_4$ emissions. For a 1 Hz sampling rate, it follows that less than 3.5 minutes of sampling in the downwind plume is needed to collect 200 in-plume measurements, offering the potential for





highly efficient sampling of site-level emissions under the right conditions. Evidently, the total flight time would need to be extended beyond 3.5 minutes to include background measurements, identify the emissions plume, and cover the necessary multiple altitudes.

Although the distribution of $CH_4$ emissions estimated through sampling the $CH_4:CO_2$ ratios at only a limited
altitude range are wider than the mass balance approach, our results show there is no consistent bias in sampling at any particular altitude over multiple days and sites. The median estimates from measured ratios in the 0-250, 250-500 and 500+ m range were all within 5% of the mass balance mean. Applying the tracer correlation method at more limited altitude ranges than the full plume, but over a greater number of days may be a more feasible strategy if full vertical coverage is not possible. Although, based on our results, this would lead to more uncertain emission
estimates than with full vertical coverage.

Since accurate wind measurements are not needed, the tracer correlation approach to estimate $CH_4$ emissions may be an appropriate measurement strategy for use with other airborne platforms such as helicopters or rotor drones where wind measurements can be affected by the rotors. We note that implementing successful measurements
would still require some logistical hurdles to be navigated. For instance, recreational drone use in Australia is limited to a maximum altitude of 120 m and authorization is needed to fly above this. It is worth noting that the tracer correlation approach could potentially be applied with platforms such as drones that are able to fly in closer proximity to estimate emissions from spatially distinct elements of a site such as power generation or the LNG trains, although the applicability for this more granular emission estimation has not been evaluated in this work,
and such surveys can present different deployment challenges and uncertainties.

In addition to lower quantification uncertainty there are several advantages to the tracer correlation approach. First, measured site-level ratios on different days could be extrapolated to reporting timescales (typically annual) without the need for detailed knowledge of $CO_2$ emissions at the time of sampling. Annual emissions of $CO_2$ are often
reported and increasingly publicly available. For example, in Australia from 2025 the regulator will publish the proportion of emissions from facilities that were $CH_4$, $CO_2$ and nitrous oxide each year. Provided the $CO_2$ emissions are reported reliably, the distribution of measured $CH_4:CO_2$ ratios could be used to estimate the distribution of annual $CH_4$ emissions. We note that extrapolating from the period of measurement to a longer timeframe such as annual assumes that the average measured emission ratios are maintained throughout the extrapolation period. Such
consistency may not be sustained at LNG facilities and this extrapolation uncertainty would need to be accounted



for. The accuracy of the longer timeframe estimate is therefore a function of the degree that $CH_4$ emissions scale with $CO_2$ (i.e. the consistency of the ratio over time). Nevertheless, provided this is taken into account, such an approach could reduce or avoid the need for access to confidential operator production or emissions data for the purposes of independent verification.


Second, given there is no need for wind measurements, and limited expertise required to calculate the $CH_4$:$CO_2$ ratio, the measurement setup could be made comparatively cheaper than an aircraft-based mass balance. Therefore, it could be repeated relatively frequently, (e.g. monthly), to ensure the accuracy of annual emission estimates. Third, the tracer correlation approach can still be applied in low or changing wind conditions where mass balance 735 may not be feasible, provided sufficient vertical coverage is achieved. That said, more work is needed to evaluate the performance of both mass balance and tracer correlation approaches in conditions such as weak, shallow sea breezes that can commonly occur around LNG facilities.

Whilst this work has focused on LNG liquefaction facilities, the $CO_2$-based tracer correlation approach to estimate 740 $CH_4$ emissions could conceivably be applied to any O&G facility with well recorded emissions of $CO_2$ that are sufficiently large. Given the precision of the LGR UGGA instrument used, $CO_2$ enhancements of a few ppm would be sufficient to retrieve useable gradients. Indeed, the smallest range of enhancements we used to estimate $CH_4$ emissions were 1-5 ppm. For a sampling distance of 3 km downwind and a $CO_2$ emission rate of around 400 t h$^{-1}$ we recorded maximum $CO_2$ enhancements of 30-35 ppm with a 5 m s$^{-1}$ wind speed. For the same instrument, this 745 suggests a source with an emission rate of 40 t h$^{-1}$ would lead to $CO_2$ enhancements 10 times larger than the measurement precision. Facilities such as offshore platforms may fall into this range. Although Shaw et al. (2023) examined $CH_4$ and $CO_2$ correlations to investigate flaring emissions from offshore platforms, they did not investigate site-level emissions. Further work is needed to test the strength of correlation between $CO_2$ and $CH_4$ from facilities such as offshore platforms, and investigate whether the $CO_2$ tracer ratio method could be used to 750 estimate site-level $CH_4$ emissions from these sites, which can have $CO_2$ emissions in the range 0.01 – 50 t h$^{-1}$ (From https://naei.energysecurity.gov.uk/data/maps/emissions-point-sources, Last access:07/01/25). For the purposes of site-level emissions reporting and verification, such as that required for OGMP 2.0, our results suggest there is the potential to exploit co-emitted $CO_2$ as a tracer of opportunity to estimate $CH_4$ emissions both accurately and efficiently via the tracer correlation method at LNG liquefaction sites.



## Data availability

Atmospheric measurement data from the Dimona aircraft will be made available via Zenodo.

## Author Contribution

MFL conceptualized the study and wrote the manuscript. Data analysis was performed by MFL, SJH and JH. JH designed the measurement campaigns, flew the aircraft, and led the data collection and processing. IJ, TR and ST collected and provided input on bottom-up data. JLF guided the research. All co-authors contributed to writing the manuscript.

## Competing interests

MFL and JLF are employees of the Environmental Defense Fund and Environmental Defense Fund Europe respectively. IJ is an employee of Woodside Energy Limited. TR and ST are employees of Chevron Australia Pty Ltd.

## Disclaimer

This publication may not represent the views of the organizations and/or companies with which the individual authors are or have been affiliated. For such views, please refer to the publications and websites of the respective organizations and/or companies.

## Acknowledgements

This research was funded in the framework of UNEP's International Methane Emissions Observatory (IMEO). The ARA Dimona research motor glider was donated by the late Joyce Schultz of Adelaide. ARA has been substantially supported by the Hackett Foundation, Adelaide. We thank Shakti Chakravarty, David Conway, Sharon Drabsch and Mei Bai for their assistance in operating the Dimona aircraft and the scientific instrumentation during the field deployments.



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
