# Peer review of "An evaluation of airborne mass balance and tracer correlation approaches to estimate site-level CH4 emissions from LNG facilities using CO2 as a tracer of opportunity"

_EGUsphere, 2025_

## Referee Comment (RC1)

This paper provides an airborne mass balance method to quantify both carbon dioxide and methane emission rates from four liquefied natural gas terminals in Australia. The authors demonstrate the performance of this mass balance method by comparing to operator-reported carbon dioxide emissions. In addition, this paper provides an alternative approach to estimate methane emission rates based on measured $CH_4:CO_2$ mole fraction compared to the previous direct mass-balance based methane measurements. However, some of the arguments regarding the feasibility of using $CH_4:CO_2$ ratio for estimating methane emissions are not well-documented and lack strong supporting discussion and should be strengthened. The current paper needs to be revised before it is publishable in EGUsphere.

Additionally, the language in this paper needs to be improved. Some of the sentences in this paper are too long, which increases the difficulty for reader to follow. Some of the sentences are ambiguously phrased, leading to potential misinterpretation and should be improved. Specific problems are outlined in the detailed comments below.

The detailed comments are summarized below:

(1) Most of the $CO_2$ emissions from the LNG terminals originate from the combustion of fuel gas or flare gas. However, the combustion efficiency is time-varying parameter related to the intermittent flaring events and combustion efficiency changes and that is difficult to quantify precisely. In practice, operators often assume a constant combustion efficiency when estimating $CO_2$ emissions, which can mask significant daily or hourly variation of $CO_2$ emissions. The approach of using measured $CH_4:CO_2$ ratio for estimating methane emissions in this study heavily depends on the accuracy of operator estimated $CO_2$ emission. The inherent inaccuracy of operator estimated $CO_2$ emissions therefore limits the reliability of using $CH_4:CO_2$ ratio for estimating site-level methane emissions.

In addition, the non-coinciding plume sources of $CH_4$ and $CO_2$ would also increase the difficulty of using tracer correlation method to estimate $CH_4$ emissions. For example, in the LNG terminals, the flare stack is usually a few hundred meters above the ground-level equipment, where fugitive methane emissions often originate. This vertical separation violates the co-emission assumption underlying the ratio method, as the $CH_4$ and $CO_2$ plumes may not be well-mixed or co-located in the sampled air mass. As a result, such condition can significantly undermine the validity of $CH_4$ estimates based on the tracer correlation approach.

Please provide additional documentation or explanation on how the study accounts for the two issues described above.

(2) The sentence in lines 50-52 and lines 60-62 are too long. It is recommended to break them into short sentences.

(3) In this study, the measurements were only conducted at a single screen downwind of each facility. However, there are usually some co-located nature or anthropogenic emission sources, such as wetlands and natural gas compressor stations, that do not belong to LNG terminals. How to make sure that there are no emission sources from the three unmeasured sides of surrounding each LNG facility entering the measured screen?

(4) For better understanding, please provide the units of parameters in all equations and figure coordinates titles. Specifically, does the site-level emission rate E in equation (1) refer to mass emission rate (e.g. kg/hr) or volume emission rate (e.g. Scf/hr)?

(5) In the first step of determining the background emission rate, why the $10^{th}$ percentile not other percentile was selected? In addition, why two times the measurement precision not one time was selected? Please provide some clarification.

(6) For Equation (3), the text in line 317 mentioned that any $CH_4$-only or $CO_2$-only plumes were ignored. Is it "$CO_2$-only plumes" or "$CO_2$-only measurements points at each transect"?

(7) For Figure 3, it is recommended to use different colors to represent different LNG terminals for better understanding the variations across different sites. In addition, there are three cluster of site-level measurements with almost constant operator estimated $CO_2$ emissions, but with one cluster operator estimated $CO_2$ emissions ranging from 600-700 t/hr. What is the reason for this discrepancy? Can you clarify it in the main text?

(8) In lines 366-368, the text shows that the 20% of relative difference between operator estimate and mass-balance measurements is impact by the monthly aggregated total emissions at two sites. I do not totally agree with this point, because the case where some mass-balance measurements are larger or smaller than operator estimate exists for every site, not only at the two sites that use monthly aggregated total emissions. The authors should dig deeper into the reasons for these larger discrepancy between measurements and operator estimates and make a clarification.

(9) In lines 450-458, the authors made a conceptual shift from the daily variation of $CO_2$ emissions measurements to operator estimated $CO_2$ emissions. The Figures 5 shows a similar random variation between $CH_4$ measurements and $CO_2$ measurements, but it does not mean the operator estimated $CO_2$ emissions and measured $CH_4$ emissions have similar random variation. I think there should be a larger daily variation than 3% in the measured $CO_2$ emissions based on the observation from Figure 3. Please make a clarification here.

(10) In Figure 7 (b)-(d), the y-axis OLS $CH_4:CO_2$ anomaly is calculated as relative to the daily mean ratio, so it only represents the daily variation. So, these figures can only represent the relationship between sampling variables and daily $CH_4:CO_2$ ratio variation, not the relationship between sampling variables and measured $CH_4:CO_2$ ratio. Please clarify it.

(11) In line 557, it should be "Fig. 8", not "Fig.7". It is recommended to add the explanations about the difference of Fig 7 (a) and (b) in the figure caption.

(12) what is the meaning of the bracket in the title of y-axis of Figure 9? Please correct it. Same problem in both the x-axis and y-axis title of Figure 10.

(13) Please correct the typo error of "$CH_4;CO_2$" in the caption of Figure 9.

(14) In line 655, it should be "Figure 10", not "Figure 9".

(15) In lines 716-718, the authors argue that one of the benefits of tracer correlation is that the measured site-level ratios on different days could be extrapolated to reporting timescales without the need for detailed knowledge of $CO_2$ emissions at the time of sampling. However, caution must be exercised when extrapolating these site-level ratios from snapshot measurements to longer timeframes, as the $CH_4:CO_2$ emission ratio is inherently time-variable. The tracer correlation measurements on a few days capture only brief snapshots of a site's emission profile. Without a comprehensive understanding of the underlying $CH_4$ and $CO_2$ emission sources, including their duration, frequency, and their magnitude, such extrapolation risks producing highly inaccurate, and potentially misleading annual emission estimate.

(16) The paragraph beginning on line 731 should be moved to the end of the above paragraph.

(17) The overall coefficients of determination shown in the Figure S4 in the supporting information are not consistent with the results shown in Figure 6 in the main text. Please correct it.

---

## Author Comment (AC1)

**Reviewer 1**

This paper provides an airborne mass balance method to quantify both carbon dioxide and methane emission rates from four liquefied natural gas terminals in Australia. The authors demonstrate the performance of this mass balance method by comparing to operator reported carbon dioxide emissions. In addition, this paper provides an alternative approach to estimate methane emission rates based on measured CH4:CO2 mole fraction compared to the previous direct mass-balance based methane measurements. However, some of the arguments regarding the feasibility of using CH4:CO2 ratio for estimating methane emissions are not well-documented and lack strong supporting discussion and should be strengthened. The current paper needs to be revised before it is publishable in EGUsphere.

Additionally, the language in this paper needs to be improved. Some of the sentences in this paper are too long, which increases the difficulty for reader to follow. Some of the sentences are ambiguously phrased, leading to potential misinterpretation and should be improved. Specific problems are outlined in the detailed comments below. The detailed comments are summarized below:

We thank the reviewer for taking the time to review the manuscript and respond in detail to their general and specific comments below. Our comments are in blue and text additions to the manuscript in red italics. Page and line numbers refer to the tracked changes version of the manuscript.

(1) Most of the CO2 emissions from the LNG terminals originate from the combustion of fuel gas or flare gas. However, the combustion efficiency is time-varying parameter related to the intermittent flaring events and combustion efficiency changes and that is difficult to quantify precisely. In practice, operators often assume a constant combustion efficiency when estimating CO2 emissions, which can mask significant daily or hourly variation of CO2 emissions. The approach of using measured CH4:CO2 ratio for estimating methane emissions in this study heavily depends on the accuracy of operator estimated CO2 emission. The inherent inaccuracy of operator estimated CO2 emissions therefore limits the reliability of using CH4:CO2 ratio for estimating site-level methane emissions.

We acknowledge the reviewer's concern about the impact of flare efficiencies on operator-reported $CO_2$ emissions. However, there is sufficient evidence presented within the paper to suggest that this concern is unfounded. Whilst the evidence for this is shown in Figures 8-10, this perhaps wasn't clear enough. There are two main reasons why a fluctuating combustion efficiency is unlikely to be detrimental to the approach:

1) Fluctuations in combustion efficiency will have only a minor impact on the accuracy of $CO_2$ emissions reporting. For flaring, emissions are calculated for $CO_2$ based on a 98% combustion efficiency following standard reporting methods. Of course, this efficiency is unlikely to always be true, and flares may vary between high efficiencies of >99% and lower than 95%. However, whilst a change in efficiency from 99% to 95% would have a large impact on $CH_4$ emissions (a 400% increase) this would only decrease $CO_2$ emissions by 4%. This error magnitude is well within the variation of the atmospheric measurement-based estimates.

2) Only a small proportion of $CO_2$ emissions from LNG terminals come from flares. Based on publicly available company GHG management plans, on average, for the facilities considered in this study, only 8% of total $CO_2$ emissions are estimated to be from flaring under normal operations (i.e. not during start-up operations).

It follows from these two points that a 10% error in flare combustion efficiency would have only a 0.8% impact on facility level $CO_2$ emissions. Given the standard deviation of mass balance $CO_2$ estimates is 25% on average, it is evident that the effect of flare combustion uncertainty is small compared to the inherent mass balance uncertainties.

Whilst operator $CO_2$ estimates may not be 100% accurate, this potential uncertainty in bottom-up $CO_2$ emission estimation is included within the manuscript both in the comparison of mass balance estimates, and by incorporating uncertainty in the operator $CO_2$ estimates into the tracer correlation calculations via a Monte Carlo sampling approach. As discussed in Section 2.4, operator estimated $CO_2$ uncertainties are estimated to be between 9-25% at the 95% confidence level based on uncertainty calculations used in the National Greenhouse and Energy Reporting Scheme.

As Fig. 9 and 10 show, the tracer correlation and mass balance estimates are within 5% of each other, which is largely explained by the 3% difference between mass balance $CO_2$ estimates and operator estimated $CO_2$. 95% of tracer correlation and mass balance $CH_4$ estimates are within +/- 40% of each other. If the operator estimates of $CO_2$ were too inaccurate to be usable as suggested by the reviewer, this result would clearly not be the case.

In response to this comment, we have added additional information on the breakdown of $CO_2$ sources within LNG terminals to the introduction text.

*P3. L86: Based on publicly available greenhouse gas management plans for LNG facilities in Australia, between 30-50% of $CO_2e$ emissions are expected from the processing trains, 10-25% from power generation and 20-40% from acid gas removal. Flaring, which may be more intermittent than other sources, accounts for less than 10% $CO_2e$ emissions on average.*

We have highlighted in the introduction that one of the purposes of this paper is to assess the applicability of the tracer correlation method.

*P5. L138: We compare the distribution of $CH_4$ emissions estimated via mass balance and tracer correlation to ascertain whether the use of $CO_2$ as a co-tracer is a reliable approach to estimate emissions.*

We have also made clear in the methods section how the Monte Carlo approach works to incorporate uncertainty in the $CO_2$ emission estimate into the tracer correlation $CH_4$ estimate.

*P14. L356: To estimate $CH_4$ emissions the ratio must be multiplied by the operator estimate of $CO_2$ emissions. For LNG facilities in Australia, these estimates are based on emission factors and energy consumption data following methods outlined in the NGER scheme. However, there is some uncertainty in these $CO_2$ emission calculations. For example, when using the simplest estimation method, the NGER measurement determination estimates a $CO_2$ emission factor uncertainty of 4%*

*for unprocessed natural gas at the 95% confidence interval. Uncertainty levels for the quantity of gaseous fuel combusted are between 1.5-7.5%.*

*To incorporate this uncertainty into the calculation of $CH_4$ emissions we use a Monte Carlo sampling approach. $CO_2$ emission values are sampled from a normal distribution, with the mean centred on the operator estimate, and standard deviation based on the estimated uncertainties outlined in section 2.4. An emissions distribution is created by sampling from both the distribution of $CH_4$:$CO_2$ ratios and the distribution of operator $CO_2$ estimates. In this work for each day of sampling we generate a 200-member distribution of $CH_4$ estimates based on sampling of the ratio distribution and $CO_2$ emissions distribution. This allows uncertainty in both the ratio calculation and the operator $CO_2$ estimate to be carried forward into the $CH_4$ estimate.*

Since it may not have been clear to the reader, we further emphasize how the comparison of mass balance and tracer correlation $CH_4$ estimates provides confidence in the use of $CO_2$ as an appropriate co-tracer.

*P34. L755: Although the exact $CO_2$ emissions may be uncertain and co-location of $CH_4$ and $CO_2$ sources may be imperfect, the results show that using tracer correlation based on co-emitted $CO_2$ results in $CH_4$ estimates that are consistent with those estimated via mass balance.*

In addition, the non-coinciding plume sources of CH4 and CO2 would also increase the difficulty of using tracer correlation method to estimate CH4 emissions. For example, in the LNG terminals, the flare stack is usually a few hundred meters above the ground-level equipment, where fugitive methane emissions often originate. This vertical separation violates the co-emission assumption underlying the ratio method, as the CH4 and CO2 plumes may not be well-mixed or co-located in the sampled air mass. As a result, such condition can significantly undermine the validity of CH4 estimates based on the tracer correlation approach. Please provide additional documentation or explanation on how the study accounts for the two issues described above.

Whilst this may be a valid concern a priori, the evidence within the paper shows it is not the case. First, the key point is that the ratios are used for site level emissions, not source level and exact co-location of emissions is not necessary. We do not presume for example that the measured ratios are representative of the individual combustion efficiencies of flares or gas turbines.

Section 3.1 shows that the mass balance approach can adequately represent the integrated mass flux from a facility. If mass balance estimates are reasonable representations of site level emissions, it follows that the ratio of integrated $CH_4$:$CO_2$ mass fluxes must be representative of the site-level emissions ratio. What we demonstrate in Section 3.6.3 is that conducting a linear regression of $CH_4$ and $CO_2$ enhancements from different altitudes achieves this same result of estimating the site-level emissions ratio, without needing to perform the integration. Section 3.6.3 along with Figure 8 are dedicated to discussing the importance of sampling across the vertical distribution to replicate the integrated mass ratio and mitigate the impacts of features such as imperfect source co-location. Figure 9 further demonstrates the representativeness of this approach relative to the ratio of fully integrated plumes.

As noted above, flares are not the dominant source of $CO_2$ at LNG sites when the processing trains are operational. Indeed, at multi-train facilities they may account for less than 5% of total

emissions. Not to mention incomplete combustion from flares is a source of methane in addition to $CO_2$. Innocenti et al (2023) estimated across 3 LNG sites that flares contributed between 5-48% total $CH_4$ emissions, depending on operating conditions and number of processing trains. It is also worth noting that they also estimated between 40-70% of $CH_4$ emissions to be from the LNG trains and power generation units. These same functional elements account for the majority of $CO_2$ emissions. In other words, it is likely that a significant proportion of $CH_4$ and $CO_2$ sources are roughly co-located.

As with the previous comment, if imperfect knowledge of $CO_2$ emissions or non-coinciding sources were a problem that precluded the use of $CO_2$ for tracer correlation it would be evidenced in the comparison between mass balance and tracer correlation estimates. The distribution of mass balance and tracer correlation estimates would not be as similar as shown in Figure 10.

In response to this comment, in Section 3.6.3 we have attempted to emphasize that for site-level emissions, exact co-location is not necessary if one is comparing the integrated plume mass of $CH_4$ and $CO_2$. We have edited the explanation of the linear regression across vertical samples as approximating this integrated plume ratio without necessarily needing to encapsulate the entire mass flux. We hope this enables the evidence to be more easily understood by the reader.

Please see the changes throughout Section 3.6.3 and 3.6.4. We have also added the following to Section 4:

*P35. L775: The most accurate way to calculate the site-level ratio is to calculate the ratio of integrated $CH_4$ and $CO_2$ mass fluxes since this implicitly accounts for different source locations and atmospheric transport errors cancel out. However, our results demonstrate there are more efficient ways to estimate this site-level emission ratio without needing to integrate the entire mass flux.*
*Through sampling random enhancement pairs across different altitudes, our results demonstrate that a simple linear regression results in mean $CH_4$ emission estimates that are consistent with mass balance estimates, with a mean difference of 5%.*

(2) The sentence in lines 50-52 and lines 60-62 are too long. It is recommended to break them into short sentences.

We have rephrased these sentences to make them easier to follow.

(3) In this study, the measurements were only conducted at a single screen downwind of each facility. However, there are usually some co-located nature or anthropogenic emission sources, such as wetlands and natural gas compressor stations, that do not belong to LNG terminals. How to make sure that there are no emission sources from the three unmeasured sides of surrounding each LNG facility entering the measured screen?

In response to this point and a comment from Reviewer 2 we have added some additional information on surrounding geography, site infrastructure and neighbouring sources. We have also added an additional figure to the supplement to show the difference between curtains that could be used for quantification and one that could not due to a neighbouring source. We also point the reviewer to P. 9 Lines 212-216 where the approach to identify other sources is outlined including

transects extending well beyond the outer edges of identified plumes and additional upwind flight paths. As noted in the manuscript, wind speeds were generally above 3 m s$^{-1}$ resulting in well-defined plumes from the LNG facilities and limiting the impact of unresolved transport processes that might introduce the type of artifacts mentioned.

(4) For better understanding, please provide the units of parameters in all equations and figure coordinates titles. Specifically, does the site-level emission rate E in equation (1) refer to mass emission rate (e.g. kg/hr) or volume emission rate (e.g. Scf/hr)?

To avoid Equation 1 becoming unwieldy, we have clarified that the calculation is the mass emission rate within the text. Units are already included in all figure axis labels.

(5) In the first step of determining the background emission rate, why the 10th percentile not other percentile was selected? In addition, why two times the measurement precision not one time was selected? Please provide some clarification.

The choice of quantile is clearly subjective, but the objective is to establish a minimum level at which there are enough data to reasonably define an average background mole fraction. Using the minimum value instead for example might introduce elevated enhancements due to an imprecise reading below the average background. Based on the flight patterns flown, we know that the transects extended beyond the plume extents and assuming 10% was a reasonable starting point. Whilst some subjectivity is inevitable, as noted in Line 217, bullet point 1, we let the data decide the number of data points to include in calculating an average background level across each transect. For all transects in this study the resulting quantile estimates used were between (12-90) with a mean of 60. i.e. on average 60% of data within each transect were within 2σ of the first decile indicating that the majority of data points were likely representative of background.

Two times the measurement precision was used to ensure that 97.6% of likely background measurements were included as background rather than 84% using 1 sigma.

We have added some clarification on both these points to the text.

(6) For Equation (3), the text in line 317 mentioned that any CH4-only or CO2-only plumes were ignored. Is it "CO2-only plumes" or "CO2-only measurements points at each transect"?
The latter. We have updated the text to clarify this.

(7) For Figure 3, it is recommended to use different colors to represent different LNG terminals for better understanding the variations across different sites. In addition, there are three cluster of site-level measurements with almost constant operator estimated CO2 emissions, but with one cluster operator estimated CO2 emissions ranging from 600-700 t/hr. What is the reason for this discrepancy? Can you clarify it in the main text?

This is explained in section 2.4 "Operator $CO_2$ emission estimation". Two sites have high resolution data. Two sites have monthly estimates. The site in question was measured over 3 separate weeks in both 2021 and 2023 which explains the variation in operator $CO_2$ estimates. That said, the variation is relatively small as a percentage of the total emissions with a standard deviation of 3 %. We have added this to the text.

*P16. L418: For sites A and B, where the comparison is to hourly mean operator estimates, the standard deviations of the bottom-up hourly data are only 1 and 3%, even though measurements were taken on 10 days in two different years.*

(8) In lines 366-368, the text shows that the 20% of relative difference between operator estimate and mass-balance measurements is impact by the monthly aggregated total emissions at two sites. I do not totally agree with this point, because the case where some mass-balance measurements are larger or smaller than operator estimate exists for every site, not only at the two sites that use monthly aggregated total emissions. The authors should dig deeper into the reasons for these larger discrepancy between measurements and operator estimates and make a clarification.

The hourly operator estimates for Sites A and B have a relatively small standard deviation, suggesting the primary driver of difference is not the bottom-up estimate. We have clarified that the main cause of variation in estimates is likely to be the uncertainty in mass balance quantification rather than operator variability. Further reasons for the differences, such as wind speed are already explored in the following sections 3.2 and 3.3. We have replaced the original text in Lines 366-368 with the following:
*P16. L418:  For sites A and B, where the comparison is to hourly mean operator estimates, the standard deviations of the bottom-up hourly data are only 1 and 3%, even though measurements were taken on 10 days in two different years. Therefore, the relative difference of individual quantifications is likely a feature of the quantification uncertainty and not emissions variability. Whilst the comparison to the other two sites is to a monthly aggregate, it is likely most of the individual quantification differences is also a feature of the mass balance quantification rather than emissions variability.*

(9) In lines 450-458, the authors made a conceptual shift from the daily variation of CO2 emissions measurements to operator estimated CO2 emissions. The Figures 5 shows a similar random variation between CH4 measurements and CO2 measurements, but it does not mean the operator estimated CO2 emissions and measured CH4 emissions have similar random variation. I think there should be a larger daily variation than 3% in the measured CO2 emissions based on the observation from Figure 3. Please make a clarification here.

It is worth clarifying here that this section does not postulate that the source level variability of $CH_4$ is the same as the 3% source level variability of $CO_2$. Rather, that the mass balance quantification uncertainty (which has a mean absolute error of 20%) is likely to be similar.

As noted above, the daily standard deviation of 3% in $CO_2$ emissions refers to the operator estimates. i.e. the actual $CO_2$ emissions across the sampling days were relatively constant. The same standard deviation exists within days. The observations in Figure 3 are based on multiple days across multiple weeks in different years. Whilst by eye it may seem like the variation is greater, the 3% standard deviation on operator estimates is correct.

Figure 5 shows there is a strong positive correlation between $CO_2$ and $CH_4$ anomalies relative to the daily mean. i.e. If $CO_2$ is overestimated by 20% then it is likely that the $CH_4$ estimate will be similarly overestimated.

We have added some additional clarification to the text.

*P20. L501: The daily standard deviation of 10-minute operator $CO_2$ estimates was 3% on average across the days of measurement. This suggests that the $CO_2$ emissions were effectively constant and the $CO_2$ anomalies each day are due to random quantification error, rather than actual emission variation. Given that similar relative magnitudes are apparent in the $CH_4$ anomalies, this implies similar random quantification uncertainties apply to $CH_4$. This assumes that the true $CH_4$ emissions over each day's sampling period of 1-4 hours were roughly constant, but it does not assume that $CH_4$ emissions were constant across different days.*

(10) In Figure 7 (b)-(d), the y-axis OLS CH4:CO2 anomaly is calculated as relative to the daily mean ratio, so it only represents the daily variation. So, these figures can only represent the relationship between sampling variables and daily CH4:CO2 ratio variation, not the relationship between sampling variables and measured CH4:CO2 ratio. Please clarify it.

We thank the reviewer for spotting this. Since the distance and wind speed variables are roughly constant each day, the plots should have been plotted as the anomaly relative to the mean ratio of each site, rather than the daily anomaly. We have revised this figure and the accompanying text. The underlying implications are unchanged.

(11) In line 557, it should be "Fig. 8", not "Fig.7". It is recommended to add the explanations about the difference of Fig 7 (a) and (b) in the figure caption.

The reference to Fig. 7 is correct. We have updated the Fig. 8 caption, which did not include an explicit explanation of (a) and (b).

(12) what is the meaning of the bracket in the title of y-axis of Figure 9? Please correct it. Same problem in both the x-axis and y-axis title of Figure 10.

The normalized quantity is unitless. The journal doesn't provide guidelines, but we've updated this to "1".

(13) Please correct the typo error of "CH4;CO2" in the caption of Figure 9.

Updated

(14) In line 655, it should be "Figure 10", not "Figure 9".

The reference to Figure 9 in this sentence was correct, but we've deleted the sentence to avoid confusion.

(15) In lines 716-718, the authors argue that one of the benefits of tracer correlation is that the measured site-level ratios on different days could be extrapolated to reporting timescales without the need for detailed knowledge of CO2 emissions at the time of sampling. However, caution must be exercised when extrapolating these site-level ratios from snapshot measurements to longer timeframes, as the CH4:CO2 emission ratio is inherently time-variable. The tracer correlation measurements on a few days capture only brief snapshots of a site's emission profile. Without a comprehensive understanding of the underlying CH4 and CO2 emission sources, including their

duration, frequency, and their magnitude, such extrapolation risks producing highly inaccurate, and potentially misleading annual emission estimate.

We note that the use of $CH_4$:$CO_2$ ratios for annual emission estimation was caveated within the second half of this same paragraph. Whilst we appreciate the point the reviewer is trying to make, the same aspersion of "highly inaccurate" could be aimed at any emission estimation method (time-limited measurement-based, continuous monitoring or applying emission factors) depending on how it is applied and without considering the available evidence. This is hardly unique to the tracer correlation approach, and we have been careful to ensure our conclusions and discussion are indeed based on evidence.

It is worth noting that LNG facilities are, by and large, continuously operating, and gas flows to the facility (and hence $CO_2$ emissions) may have relatively limited variation across the year, provided there are no major maintenance events, planned or unplanned. For example, records from the Australia Energy Market Operator (AEMO) show that the standard deviation of daily pipeline flows (used as a proxy for total site activity and hence $CO_2$ emissions) to one LNG facility within the 2024 calendar year was 9%. We note that there is some proportionality between LNG production rates and ambient temperatures with air-cooled, gas turbine-driven facilities and so some variability is expected.

The point being conveyed is that one does not necessarily need to have access to the *exact* $CO_2$ emissions from the time of measurement (which may not be available or known). Provided one has knowledge such as the above, it is possible to make reasonable assessments of the probability of inaccurate interpretations. Of course, one should take into account potential factors that might affect the measurements (such as turnarounds, unusual flare activity, pipeline flow data if available). An important consideration is to build up a distribution of $CH_4$:$CO_2$ ratios that can be used to better represent how ratios vary across a year from which one can build up a representative measurement-based annual estimate.

In light of the reviewer's comment, we have edited this paragraph to clarify that additional information is useful to interpret the ratios and that, clearly, a larger distribution of samples will lead to more representative mean and uncertainty estimates. We note that a follow-up study on the use of tracer correlation for annual emissions verification is currently in prep by some of the same authors.

*P37. L817: In addition to lower quantification uncertainty there are several advantages to the tracer correlation approach. First, measured site-level ratios on different days may be a more reliable indicator of emissions across reporting timescales (typically annual) than a simple temporal extrapolation of a $CH_4$ emissions estimate alone. This is because the ratio inherently incorporates changes in activity data through the $CO_2$ enhancement denominator. Whilst this may reduce the need for exact knowledge of $CO_2$ emissions at the time of sampling there are some important caveats. Some knowledge of operating conditions (such as the occurrence of maintenance events) remains important to help evaluate if the measured ratios are likely to be representative of normal operating conditions. For operators implementing this approach themselves, this knowledge would be a given. However, even for external validation some useful information is often available. For example, in Australia, operators are generally required to publish major maintenance events (processing train turnarounds) in advance, port records provide information on the regularity of*

*exports and additional data on pipeline flows to certain facilities are available via the Australian Energy Market Operator.*

*Additionally, it is important to build a distribution of site-level ratio estimates to better understand the likely variability in emission ratios across longer time periods and reduce uncertainty in the extrapolation to annual emissions. The accuracy of the longer timeframe estimate is a function of the degree that $CH_4$ emissions scale with $CO_2$ (i.e. the consistency of the ratio over time) and a larger number of samples will inherently reduce the associated uncertainty. Provided these factors are considered, such an approach could reduce or avoid the need for access to confidential operator production or emissions data for the purposes of independent verification.*

(16) The paragraph beginning on line 731 should be moved to the end of the above paragraph.

The additions above have already necessitated changes to the paragraph structure.

(17) The overall coefficients of determination shown in the Figure S4 in the supporting information are not consistent with the results shown in Figure 6 in the main text. Please correct it.

Updated in the supplement.

**References**

Innocenti, F., Robinson, R., Gardiner, T., Howes, N., and Yarrow, N.: Comparative Assessment of Methane Emissions from Onshore LNG Facilities Measured Using Differential Absorption Lidar, Environ. Sci. Technol., 57, 3301–3310, https://doi.org/10.1021/acs.est.2c05446, 2023.

---

## Author Comment (AC2)

**Reviewer 2 comments**

Interesting study, focusing on the emission quantification of LNG facilities. Especially the use of the already emitted CO2 as a tracer is valuable. This study is therefore a good addition within this field of research. It also gives a good overview of the advantages and disadvantages of the mass balance and tracer approach. Furthermore, the statistical substantiation is well provided. The paper is written clear and a relevant topic, that contributes to a better understanding of methane emissions in the environment.  It is also helpful to the OGMP 2.0 implementation as a potential method.

We thank the reviewer for their comments and address the specific issues below in turn. Our responses are in blue and additions to the manuscript text in red italics. Page and line numbers refer to the tracked changes version of the manuscript.

**Specific comments:**

In the abstract, the authors present 2 approaches, downwind mass balance method (1) and the tracer correlation approach (2). In line 38 the authors mention the tracer correlation approach can be used, but here the mass balance approach is not mentioned. Maybe good to also mention the results of this approach in comparison to approach 2.

We have added the mass balance approach to this statement on the similarity of the results. The sentence now reads:
*P2. L38: Our results indicate that the mass balance and the $CO_2$-based tracer correlation approaches provide estimates of $CH_4$ emissions from LNG facilities that are consistent with each other.*

I miss information on the site description and size of the LNG facilities. I understand that they are anonymised, but later on some measurements are above water instead of land. This can have major influences on the dispersion and mixing of the plumes.

As noted, the sites are deliberately anonymized, partly to allow the focus to be purely on the methods. However, we acknowledge that certain aspects may be a little too implicit and require prior knowledge of typical LNG features. Therefore, we have added some further information on the location and size of the four sites alongside the infrastructure that this study covers to the methods section 2.2.

*P7. L186: Each of the four LNG facilities is located within close proximity to the coastline, allowing easy access for LNG ships to transport the LNG to customers. The facilities range in size from a single processing train to multiple trains which impacts the amount of LNG produced and the expected emissions. Each processing train is around 200-300 m in length and each facility covers an area of at least 1 $km^2$, not including the transfer jetty. In addition to processing trains, sites include power generation units, multiple flares, storage tanks, domestic gas processing, condensate processing and other infrastructure. Given the size and potential for multiple sources*

*of both CH$_4$ and CO$_2$, measurements used for quantifications were performed at a downwind distance of between 2-17 km from the centre of each site.*

*Each LNG facility is situated in a sparsely populated area of Australia. Potential neighbouring CH$_4$ and CO$_2$ sources included one of the other LNG facilities, a neighbouring industrial chemical facility, neighbouring O&G infrastructure and termite mounds. To ensure the influence of these were excluded the following strategy was deployed. For each site a reconnaissance leg was flown to identify detectable sources, including any upwind of the LNG facilities. Where another non-LNG plume was detected, only those curtains that had clearly defined separable plumes were used in the analysis. An example is shown in SI Fig S1. Whilst low-level dispersed emissions may have been generated by sources such as termite mounds, these were not detectable as plumes and any widespread upwind contribution of these would be incorporated into the background component which is removed to calculate the enhancement attributable to LNG emissions.*

Line 488: For completeness, maybe add that also because of closer distances, plumes from differenced sources of the facilities will not be well mixed already. At closer distances, the individual sources can be detected, which may result in difference CH4:CO2 ratios. This is also shown in Figure 6, with the different sources having different CH4:CO2 ratios at closer distances (Figure 6 a, b and c) and are well mixed at further distances (8 km; Figure 6 d).

Agreed, we have added this point to the text.
*P22. L542: Furthermore, at closer distances the plumes from individual sources are unlikely to have mixed and transect ratios may be representative of individual sources, rather than the site-level average. It is notable in Fig.6 that the most consistent site-level correlation is achieved from the measurements taken furthest away, shown in Fig 6(d).*

Line 538: Here the authors mentioned that measurements were often taken over water. This can also explain part of the poor mixing below 250 m (line 536). The mixing/dispersion of plumes behave different above water compared to above land (this is for example included in the EPA OCD (Offshore and Coastal Dispersion) model.

Thank you for pointing this out. We have added this point to the discussion.
*P27. L595: Since the measurements were often taken over water, other considerations could be the presence of LNG tankers or other ships that might have caused these larger variations as well as poorer mixing due to reduced surface heating over water than over land.*